# A review and *in silico* screening of plant-derived snake venom/toxin inhibitors: ADMET, drug-likeness, and medicinal chemistry profiling

Prince Ojuka[1], George S. Nyamato[1], Cleydson B.R. Santos[2,3], Njogu M. Kimani ● [1,4]*

1 Department of Physical Sciences, University of Embu, Embu Kenya, 2 Laboratory of Modeling and Computational Chemistry, Department of Biological Sciences and Health, Federal University of Amapá, Macapá, Amapá, Brazil, 3 Graduate Program in Medicinal Chemistry and Molecular Modelling, Health Science Institute, Federal University of Pará, Belém, Pará, Brazil, 4 Natural Product Chemistry and Computational Drug Discovery Laboratory, Embu, Kenya

* njogu.mark@embuni.ac.ke

## Abstract

Snakebite envenomation remains a pressing global health issue, contributing to high rates of illness and death. It also imposes significant socio-economic burdens on affected communities. Recent conservative estimates suggest that approximately 5.4 million snakebite incidents occur annually, leading to nearly 138,000 fatalities. In Africa alone, as many as 500,000 cases are documented each year. This research seeks to explore phytochemicals reported to exhibit *in vitro* and *in vivo* inhibitory effects on snake venom and toxin targets. A systematic review was conducted utilizing six electronic databases for literature searches, including Google Scholar, Web of Science, ScienceDirect, Scopus, Springer, and PubMed. This process identified 213 phytocompounds with inhibitory activity against the specified targets. Computational tools such as SwissADME, pkCSM, ADMETlab, ProTox3, Toxtree, and DataWarrior were employed to assess the absorption, distribution, metabolism, excretion, and toxicity (ADMET) characteristics, alongside other medicinal chemistry properties of these compounds. The results indicate that several plant-derived molecules effectively inhibit snake venom/toxin targets *in vitro* and *in vivo*; however, only a few appear suitable for drug development without further modifications. Among the analyzed compounds, the terpenes labdane lactone and labdane trialdehyde, along with the benzenoid anisic acid, exhibited strong antivenom potential. Notably, anisic acid achieved complete (100%) neutralization of lethality and defibrinogenation induced by the venoms of *Naja kaouthia*, *Daboia russelii*, *Ophiophagus hannah*, and *Echis carinatus* in both *in vivo* and *in vitro* studies. Labdane lactone and labdane trialdehyde, which were isolated from *Curcuma antinaia* and *Curcuma zedoaroides*, respectively, demonstrated significant venom-inhibitory activity at a concentration of 22.7 µM and 21.9 µM against *Ophiophagus hannah* venom. Specifically, labdane lactone exhibited an inhibition rate of 83%, while labdane trialdehyde achieved a 62%

**Data availability statement:** All relevant data are within the manuscript and its Supporting information files.

**Funding:** The author(s) received no specific funding for this work.

**Competing interests:** The authors have declared that no competing interests exist.

inhibition rate against the venom of *Ophiophagus hannah* along with favorable *in silico* drug-likeness and ADMET profiles.

## Author summary

Snakebite remains a critical global health concern, particularly in Africa, where half a million cases occur annually. The search for effective, affordable antivenoms has driven interest in plant-based compounds with inhibitory effects on venom toxins. In this study, we reviewed and analyzed 213 phytochemicals reported to neutralize snake venom activities *in vitro* and *in vivo*. Using a combination of six databases and computational tools for drug-likeness and toxicity profiling, we identified promising candidates with favorable pharmacological properties. Notably, three compounds anisic acid, labdane lactone, and labdane trialdehyde exhibited strong antivenom potential. Anisic acid completely neutralized lethal and defibrinogenating effects of four medically significant snake venoms, while labdane derivatives from *Curcuma* species showed high inhibition rates against *Ophiophagus hannah* venom. These findings suggest that a small number of phytochemicals possess both potent bioactivity and favorable ADMET profiles, supporting their further development as potential adjunct or alternative snakebite therapies.

## 1. Introduction

Snakebite envenomations (SBEs) are a significant but often neglected public health concern, affecting populations globally, particularly in tropical and subtropical developing regions [1,2]. High cases in these areas are partly attributed to the preservation of forests and biodiversity. However, human expansion and urbanization into these biodiverse regions have led to increased human-snake interactions, particularly in Latin America, Sub-Saharan Africa, and Asia [2–4]. Annually, an estimated over five million snakebites occur worldwide, resulting in approximately 138,000 deaths and leaving around 400,000 individuals with lasting physical disabilities [5,6]. Asia reports the highest number of cases, accounting for 73% of global incidents (two million), with India alone recording over forty six thousand deaths in 2020 [1]. Africa and the Middle East follow, with about five hundred and eighty thousand cases (21%), including seven thousand to thirty two thousand deaths in Sub-Saharan Africa [1,7]. Latin America and the Caribbean report approximately one hundred and fifty thousand cases (5%) and five thousand deaths, with most incidents occurring in South America. Brazil encounters twenty six thousand to twenty nine thousand cases annually, roughly one-third of which take place in the Amazon region [6,8].

These estimates likely underrepresent the true number of SBE cases, as many incidents go unreported [4]. Underreporting is common because SBEs often occur in remote rural areas where access to healthcare services is limited [4,9]. Despite

significant advancements in healthcare, managing snakebites remains a challenge. While antivenom therapy can reduce mortality, it does not address local tissue damage [10]. Furthermore, the availability of antivenom is scarce in many remote regions [2]. Due to these treatment challenges, some communities have developed traditional methods to mitigate snakebite effects [8,9]. Traditional healers worldwide use medicinal plants for treating snakebites, and it is common in these regions to use extracts or teas made from leaves, roots, or bark [1,4,8,9]. However, the unscientific and unsafe application of such remedies often leads to adverse clinical outcomes, delaying or complicating effective treatment by healthcare professionals [1].

Currently, numerous medicinal plants are being assessed through *in vivo* and *in vitro* studies to validate their therapeutic benefits. Previous phytochemical analyses of bioactive plant extracts have led to the identification of several bioactive compounds, including terpenoids, alkaloids, steroids, flavonoids, and quinones [11]. Furthermore, secondary metabolites from plants that were not previously recognized for their pharmacological potential are now being explored as sources of new medicinal agents. These bioactive compounds serve as valuable lead molecules in the process of discovering and developing new drugs [12]. Various computational approaches have been applied across different stages of discovering and developing new drugs. In recent years, *in silico* ADMET modeling has gained considerable recognition as a valuable tool in rational drug design [13–15]. The effectiveness and affordability of these models have enhanced the drug development process by enabling the prediction of compound bioavailability, aiding in hit identification, and optimizing molecular structures [16]. Recently, there has been a notable rise in research articles examining the *in silico* evaluation of plant-derived compounds. Additionally, computational screening of natural product databases has expanded rapidly [17]. However, systematic reviews specifically addressing the molecular targets of plant metabolites with anti-snake venom properties have been scarce. Although some recent reviews have examined plant-based antivenom activities, most lack detailed discussions on the underlying mechanisms, pharmacokinetics, and pharmacodynamics [1]. Another study reviewed plant-based inhibitors but focused solely on bothropic venom [18]. Unlike such compilations, this study offers a comprehensive and critical evaluation of bioactive plant-derived compounds for their therapeutic potential against snakebite envenomation. Additionally, it explores new possibilities for their application in the development of novel snakebite treatments through ADMET and medicinal chemistry profiling of plant compounds as snake toxin inhibitors.

## 2. Methods

### 2.1. Study design

This systematic review was conducted in accordance with the PRISMA (Preferred Reporting Items for Systematic Reviews and Meta-Analyses) guidelines [19]. The research question was structured using the PICO framework (Problem, Intervention, Control, Outcome), as recommended by PRISMA: "In *in vitro* and *in vivo* studies (I) using venom-related target models, what is the potential of plant-derived compounds (P) in neutralizing venom activity (O) compared to control groups and standard antivenom treatments (C)?"

### 2.2. Search strategy

An extensive review of the literature from various scientific sources, including Google Scholar, ScienceDirect, Web of Science, Scopus, Springer, and PubMed was conducted. The study databases comprised original research articles from peer-reviewed journals, books, dissertations, theses, and patents. Scientific data written in or translated into English and published up to December 2024 were included in the review. The search strategy employed combinations of keywords and Boolean operators such as: "snakebites," "snake envenomation," "snake AND venom," "natural inhibitors," "antivenom AND activity," "toxins," "plants," "phospholipase inhibitors," and "metalloprotease inhibitors," either individually or in combination. Studies presenting bioactivity data of plant-derived compounds tested in both *in vitro* and *in vivo* models against snake venoms, their specific toxins, or the biological effects induced by those toxins were considered relevant and included.

## 2.3. Inclusion criteria

Studies reporting data on the efficacy of plant-derived compounds in neutralizing snake venom through *in vitro* and *in vivo* experiments were considered for inclusion.

## 2.4. Exclusion criteria

Studies focusing solely on compounds from non-plant sources were excluded. Additionally, studies evaluating the activity of crude extracts, fractions, or compound mixtures were not considered.

## 2.5. Data collection and extraction

To organize the materials for this systematic review, we compiled all relevant articles. Duplicates were removed, and a new file containing all the relevant material and its consolidated data was generated. Data extraction included various factors: the name of each plant-derived compound, its chemical structure (which was drawn using ChemDraw v.12), the chemical class of the compound, the target venom models investigated, and the specific *in vitro* or *in vivo* assays employed. Additionally, information on the concentrations of plant compounds tested, numerical results of inhibitory activity, and units of measurement for both concentration and assay results was recorded. We also noted the types of snakes used in the studies and any observed suppression of venom activity. This extracted information underwent a thorough verification process to ensure accuracy and completeness. Any inconsistencies were addressed through discussion and resolved by consensus, with arbitration used when necessary.

## 2.6. Risk of bias

This study identified several potential sources of bias, such as the criteria for inclusion and exclusion, the selection of databases, language barriers, and variations among the studies. The quality of the studies was evaluated based on their publication in peer-reviewed journals. The quality of the studies was evaluated based on the significance of the results obtained, rather than solely on their publication in peer-reviewed journals. To minimize bias, protocols such as PICO and PRISMA were rigorously adhered to. The biases considered included factors like the inclusion/exclusion criteria, missing data, database selection, language differences, publication dates, and the effect of missing primary data. Additionally, the types of articles included in the review were assessed to maintain methodological rigor.

## 2.7. Methodology summary

We calculated all numerical data collected during the review process. This involved determining the number of articles retrieved from each database using specific search terms, the total number of articles in the compiled file, and those removed due to duplication. We also documented the number of articles that were either included or excluded based on independent evaluations, as depicted in Fig 1.

## 2.8. *In silico* analysis

To assess ADMET properties, drug-likeness, and other medicinal chemistry parameters, a combination of computational tools was utilized, including SwissADME (http://www.swissadme.ch/), pkCSM (http://biosig.unimelb.edu.au/pkcsm/), ADMETlab (http://admet.scbdd.com/), ProTox3 (https://tox-new.charite.de/protox_III/), Toxtree (https://toxtree.sourceforge.net/), and DataWarrior (http://www.openmolecules.org/datawarrior/). Molecular structures were generated, validated, and saved in SMILES format using ChemDraw v. 12 (https://revvitysignals.com/chemdraw) before being submitted to the selected software and platforms for analysis. Each parameter, including drug-likeness (DL), absorption (A), distribution (D), metabolism (M), excretion (E), toxicity (T), and medicinal chemistry (MC) was evaluated using at least two different platforms. The obtained data were cross-verified, and only the parameters with matching results were considered in the

PLOS Neglected Tropical Diseases

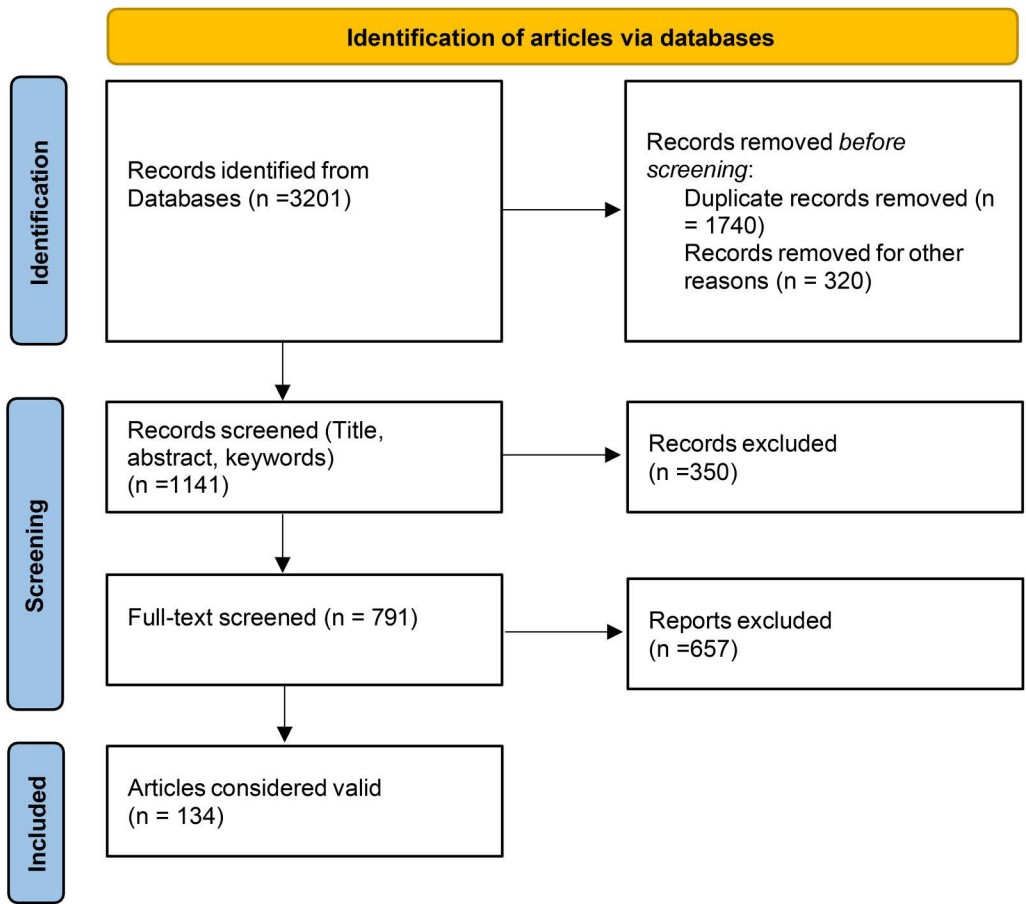

**Fig 1. PRISMA flow chart of the screening process.**

screening process to enhance the accuracy and reliability of predictions. A sequential filtering approach was applied, progressively reducing the number of compounds based on predefined cutoff criteria [20–22]. This refinement process enabled the selection of molecules with the highest pharmacokinetic and pharmacological potential.

## 3. Results and discussion

After extracting data from the reviewed and validated studies, a total of 213 plant-derived compounds were identified as inhibitors of snake toxins and venom. The analysis revealed 11 distinct chemical classes (Fig 2), with terpenoids [23], flavonoids [24], benzenoids [9], alkaloids [17], isoflavonoids [10], hydroxycinnamic acids [8], saponins [10], modified glyco-sides [8], polyketides [4], tannins [5], and coumarins [3]. Additionally, other identified compounds were 70.

### 3.1. Venom toxins

Over 250,000 species utilize venoms for various purposes, including capturing prey, deterring rivals, or defending against predators. Their evolutionary success is demonstrated by the presence of venomous organisms in every eco-system [25]. In animals like snakes, envenomation involves delivering venom into another organism's tissue through specialized fangs. This venom, a toxin-rich secretion, aids in immobilizing and digesting prey, although it also serves as a defense mechanism [1,26]. Genetic mutations and natural selection have driven the diversification of the venom

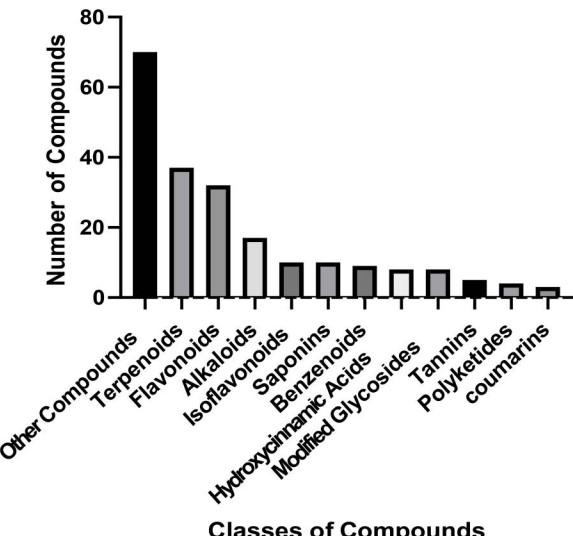

**Fig 2. Classes of compounds, and the number of compounds tested against targets of snake venom and toxins.**

proteome across snake families, resulting in species-specific toxicities [27]. Snake venoms mainly consist of proteins and peptides, making up 90–95% of their composition, which may or may not exhibit enzymatic activity. Key constituents include metalloproteases (SVMPs), phospholipases $A_2$ (PLA$_2$s), serine proteases (SVSPs), L-amino acid oxidases (LAAOs), hyaluronidases (HAases), phosphodiesterases (PDEs), acetylcholinesterases (AchEs), nucleases, disintegrins, three-finger toxins (3-FTxs), C-type lectins (CTLs), and cysteine-rich secretory proteins [22]. Not all venoms contain every toxin type, as synthesis and secretion vary across production cycles, resulting in compositional differences [28,29].

Despite containing over 20 protein families, most snake venom effects stem from four primary families: PLA$_2$s, SVMPs, SVSPs, and 3-FTxs. These proteins target multiple physiological systems, causing various pathological effects [1,30]. Elapid venoms, rich in neurotoxic PLA$_2$s and 3-FTxs, primarily induce neurotoxicity, which can escalate to respiratory failure, a potentially fatal condition. They typically cause minimal local effects [6]. Conversely, viper venoms are rich in proteases, leading to local and systemic effects, including pain, swelling, blistering, bleeding, and ecchymosis. Severe cases may result in complications such as bacterial infections, compartment syndrome, tissue necrosis, and amputations. Systemic effects like acute kidney injury are also commonly observed [6,31]. The composition and concentration of venom toxins can vary based on the snake's age, sex, diet, and geographic location, influencing the severity and type of envenomation outcomes [32,33].

**3.1.1. Phospholipases A$_2$ (PLA$_2$s).** Phospholipase $A_2$ (PLA$_2$) enzymes are found across a wide range of taxa, including bacteria, plants, invertebrates like arachnids and insects, and vertebrates, including mammalian tissues and snake venoms [1]. These enzymes belong to subgroup II of secreted phospholipases, which are categorized into two classes: Asp49 PLA$_2$s, containing aspartate at position 49, are catalytically active and hydrolyze the sn-2 acyl bond in phospholipids and free lipids in various cells; and Lys49 PLA$_2$s, containing lysine at position 49, which lack catalytic activity but exhibit significant myotoxic effects [24]. Snake venom PLA$_2$s demonstrate a broad range of biological activities, functioning individually or synergistically. These include neurotoxicity, myotoxicity, cardiotoxicity, anticoagulant effects, spasms, platelet aggregation inhibition, hypotension, and inflammation [29,34]. Despite their lack of enzymatic activity, Lys49 PLA$_2$s are highly myotoxic and play a significant role in venom-induced tissue damage. Notable examples include BaPLA$_2$I and BaPLA$_2$III, which are PLA$_2$ toxins isolated from the venom of *Bothrops atrox* [1].

**3.1.2. Snake venom metalloproteases (SVMPs).** Snake venom metalloproteinases (SVMPs) are enzymes with a molecular mass ranging from 20 to 100 kDa and are dependent on divalent metal ions, such as zinc, which are crucial for maintaining their three-dimensional structure and enabling their catalytic functions [1]. The catalytic site of SVMPs is highly conserved and includes a zinc-binding domain, HEXXHXXGXXH, and a methionine-turn motif. SVMPs are categorized into different groups and subgroups based on their domain organization: P-I contains only the metalloproteinase domain, P-II includes the metalloproteinase domain followed by disintegrin, and P-III has the metalloproteinase, disintegrin-like, cysteine-rich, and possibly a lectin-like domain, classified as PIIId [35].

Due to their size, SVMPs primarily act within the circulatory system, helping to spread the venom's toxicity and amplify the effects of smaller venom components [36]. These enzymes are key contributors to the mechanisms of envenomation, including local and systemic hemorrhage, necrosis, blister formation, and inflammation. Their proteolytic activity, facilitated by the protease domain and associated domains such as disintegrin/disintegrin-like and lectin-like, leads to the breakdown of important components of the basal membranes (e.g., laminin, fibronectin, type IV collagen), which weakens the structural integrity of blood vessels, causing hemorrhage [37]. The disruption of connective tissue also results in blister formation at the site of the bite [23,38].

Additionally, the proteolytic action of SVMPs contributes to systemic effects, such as coagulation disorders, by cleaving coagulation factors and inducing a procoagulant state. Their effect on hemostasis results in consumption coagulopathy, leading to both local and systemic bleeding [1,39]. SVMPs also stimulate leukocytes directly, acting as venom-associated molecular patterns (VAMPs) that influence the inflammatory complement system, as well as indirectly through the production of damage-associated molecular patterns (DAMPs) via proteolytic action on cellular and extracellular components [40,41].

**3.1.3. Snake venom serine proteases (SVSPs).** Similar to SVMPs, SVSPs also function as procoagulants by targeting one or more coagulation factors in the blood coagulation cascade. This contributes to the digestion of prey and impacts coagulation, fibrinolysis, platelet aggregation, and the complement system, as well as other immune responses [1,35,42]. SVSPs have molecular masses ranging from 26 to 67 kDa and contain a conserved catalytic domain with a triad of His57, Asp102, and Ser195, which is responsible for cleaving peptide bonds at arginine or lysine residues at the C-terminal [42]. By acting on the hemostatic system, SVSPs cause systemic hemodynamic disturbances, activating key proteins in the coagulation cascade and influencing systems such as the kallikrein-kinin and complement systems, thus affecting endothelial and platelet cells [42,43].

**3.1.4. Three-finger toxins (3-FTxs).** Three-Finger Toxins (3-FTxs) are a family of highly conserved non-enzymatic peptides, ranging from 60 to 74 amino acids, with various biological functions [1,29]. Proteomic and transcriptomic studies have shown that 3-FTxs are predominantly found in the venoms of Elapidae snakes, such as *Ophiophagus*, *Dendroaspis*, and *Naja* species from Africa and Asia, *Micrurus* from the Americas, as well as in venoms of *Columbridae* and *Hydrophiidae* snakes [44,45]. Most subgroups of 3-FTxs are neurotoxins that primarily target the cholinergic system and exhibit selectivity for various receptor subtypes. They also disrupt immune system regulation and inflammatory signaling through different intracellular pathways [46]. These toxins are mainly neuromuscular blockers and potent antagonists of nicotinic acetylcholine receptors (nAChRs) at the neuromuscular junction [23], contributing to cardiorespiratory failure in snakebite victims. Bungarotoxin, isolated from *Bungarus multicinctus*, is a prominent example of a potent 3-FTx, known for its strong binding to nAChRs, and is commonly used in receptor studies [46]. Cardiotoxins (CTXs), the second largest group of 3-FTxs, are also cytotoxins due to their ability to lyse various cells [46]. CTXs are acetylcholinesterase inhibitors, blocking the enzyme at neuromuscular junctions and causing muscle spasms [46,47]. In addition to blocking nAChRs, 3-FTxs are involved in several other biological responses, including inhibition of platelet aggregation, modulation of adrenoreceptors, L-type calcium channel blockade, and anticoagulant activity [48].

**3.1.5. Other toxins.** In addition to the more prevalent proteins in snake venoms, there are also less abundant components with lower structural variability. These components either enhance or complement the effects of the more

abundant toxins, while also interacting with specific physiological targets [49]. This group includes LAAOs, HAases, nucleotidases (NUCs), and PDEs. LAAOs are found in various organisms and are not exclusive to Viperidae and Elapidae snake venoms [39]. The concentration of LAAOs varies between snake species, influencing their toxicity levels. These enzymes are characterized by their catalytic specificity for hydrophobic and aromatic amino acid chains and exhibit structural and functional variability, which can disrupt platelet function, impair plasma coagulation, or lead to the death of various cell types [50]. HAases, although present in small amounts, play a role in enhancing venom diffusion in prey tissues by hydrolyzing hyaluronic acid, a key component of connective tissue, thus amplifying the effects of other toxins [1]. NUCs primarily generate adenosine through their catalytic activity, which aids in immobilizing prey by increasing vascular permeability, and also induces hypotension, bradycardia, and decreased locomotor activity. Among nucleases, PDE exonucleases are responsible for hydrolyzing phosphodiester bonds, known to induce hypotension, depress locomotor activity, and inhibit platelet aggregation [51]. Both NUCs and PDEs cleave nucleic acid derivatives and substrates like ATP, ADP, and AMP. These enzymes may act independently or work in synergy with other toxins, such as SVMPs, $PLA_2$s, and disintegrins, to inhibit platelet aggregation and enhance the anticoagulant effects of snake venoms [1,52].

### 3.2. Plant compounds as antivenom agents

Natural compound-based binding inhibitors offer a promising and safe treatment approach for snakebite envenomation. As a result, these plant-derived therapies could serve as effective alternatives, especially in cases where specific antivenom treatments are unavailable [1]. This review discusses the latest research findings on compounds with anti-snake venom properties and provides a summary of their specific molecular targets (S1 Table and S1 Fig).

**3.2.1. Alkaloids.** Alkaloids exhibit a wide range of biological activities, yet only a limited number of studies have reported their inhibitory effects on snake venom enzymes (S1 Table and S1 Fig). One such compound, schumanniofoside (**1**) , a glycosylated benzopyranylpiperidinone alkaloid derived from the stem bark of *Schumanniophyton magnificum* (Rubiaceae) in Nigeria, demonstrated the ability to reduce the lethal effects of *Naja melanoleuca* venom *in vivo* at varying doses (10–100 mg/kg), with an 80 mg/kg dosage decreasing mouse mortality by 15% [53].

A different study emphasized the *in vivo* potential of 12-methoxy-4-methylvoachalotine (**2**), an indolic alkaloid featuring an iboga-type skeleton. Extracted from the root bark of *Tabernaemontana catharinensis* (Apocynaceae) in São Paulo, Brazil, this compound fully counteracted the lethal effects of *Crotalus durissus* venom when administered at a dose of 1.7 mg/100g, 20 seconds after injecting mice with $^{two\ times}$ mean lethal doses ($LD_{50}$) [54]. Furthermore, two steroidal alkaloids, 22α,23α-epoxy-solanida-1,4,9-trien-3-one (**3**) and 22α,23α-epoxy-solanida-1,4-dien-3-one (**4**), extracted from *Solanum campaniforme* leaves (Solanaceae) in Ceará, Brazil, were evaluated for their activity against *Bothrops pauloensis* venom [55].

Their anti-myotoxic, anti-hemorrhagic, and anti-necrotizing properties were evaluated following *in vitro* venom incubation. The presence of these alkaloids significantly reduced necrotic areas (~27.0 and 32.0 mm², respectively) and minimized hemorrhagic damage [56]. Further investigation into the same plant species led to the identification of four steroidal alkaloids exhibiting biological activity against *B. pauloensis* venom, including: (E)-N-[8′(4-hydroxyphenyl)ethyl]-22α,23α-epoxy-solanida-1,4,9-trien-3-imine (**8**), 22β,23β-epoxy-solanida-4-en-3-one (**7**), 22α,23α-epoxy-solanida-4-en-3-one (**6**), (Z)-N-[8′(4-hydroxyphenyl)ethyl]-22α,23α-epoxy-solanida-1,4-dien-3-imine (**9**), and 22β,23β-epoxy-solanida-1,4-dien-3-one (**5**). Compounds **3**, **5**, **6**, and **9** demonstrated anti-hemorrhagic activity, whereas compounds **6**, **7**, **8**, and **9** exhibited anti-necrotic properties [57].

Berberine (**10**), derived from *Cardiospermum halicacabum* (Sapindaceae) in India, was identified as a phospholipase $A_2$ ($PLA_2$) inhibitor in *Daboia russelii* venom. The compound's inhibitory activity was confirmed through co-crystallization experiments, which revealed its potential as an anti-inflammatory agent for snakebite treatment [58].

Another potential antivenom agent is aristolochic acid (**11**), a nitrophenanthrene carboxylic acid found in *Aristolochia* species (Aristolochiaceae) from India. It demonstrated *in vitro* inhibitory effects against L-amino acid oxidase (LAAO) in *D.*

*russelii* venom (19% inhibition, $IC_{50} = 33.6$ μM) [59] and suppressed the venom-induced edema by 50% at a concentration of 0.16 μM [60]. Additionally, pre-incubation with this compound (13.7 μg/mL) significantly decreased the myotoxic effects of piratoxin-I (PrTX-I) from *Bothrops pirajai* venom [61]. However, aristolochic acid is known to interact with DNA, leading to carcinogenic effects, rendering it unsuitable for therapeutic use [59].

To overcome this limitation, semi-synthetic derivatives of aristolochic acid were developed by substituting the nitro group with either a chlorine atom (**12**) or a hydroxyl group (**13**). These modifications eliminated DNA binding while preserving LAAO inhibitory activity in *D. russelii* venom. Furthermore, compound **12** demonstrated superior inhibition of LAAO-induced reactive oxygen species (ROS) production in human embryonic kidney (HEK293) cells (76%) and human hepatocellular carcinoma (HepG2) cells (68%), while compound **12** also significantly reduced ROS levels [59]. 12-Methoxy-Nb-methyl-voachalotine (**13**), extracted from *Tabernaemontana catharinensis* (Apocynaceae), exhibited strong inhibitory effects against the lethality and myotoxic activities induced by *Crotalus durissus terrificus* venom. This compound effectively neutralized crotoxin, the primary toxin in the venom [62]. Shenoy et al., identified piperine (**14**) as a constituent of the ethanolic fruit extract of *Piper longum L.* Piperine was found to interact with phospholipase $A_2$ from *Daboia russelii* venom [63]. Atropine (**15**), a well-known tropane alkaloid, was found to exhibit potential antivenom activity. Its pharmacological effects, including modulation of acetylcholine receptors and inhibition of venom-induced neuromuscular blockade, contribute to its protective role in envenomation scenarios. These properties suggest that atropine could be a valuable adjunct in snakebite treatment strategies [64]. Ajmaline (**16**), known for its pharmacological activities, has demonstrated partial inhibition of *Naja naja* venom hyaluronidase (HAase) and NNH1 activity. Although not a complete inhibitor, its observed effects suggest that it may be beneficial in reducing venom spread through enzymatic neutralization [65]. Similarly, anisodamine (**17**) from *Anisodus tanguticus* (*Zang Qie*) relieved venom-induced microcirculatory issues and associated renal failure [62].

### 3.2.2. Benzenoids.

Several simple compounds containing substituted benzene nuclei have been identified in certain plants and may serve as potential inhibitors of snake venom (S1 Table and S1 Fig). One such plant, *Hemidesmus indicus* (Apocynaceae), native to India, has been the subject of studies investigating the venom-neutralizing potential of 2-hydroxy-4-methoxybenzoic acid (**18**). This compound demonstrated inhibitory effects against the venoms of *Daboia russelii*, *Naja kaouthia*, *Ophiophagus hannah*, and *Echis carinatus* [66–68]. It effectively counteracted the venom's toxic effects, reducing lethality, hemorrhagic activity, and coagulation disturbances, while also mitigating the inflammation induced by *D. russelii* venom in rodents.

An additional compound derived from the same plant, 2-hydroxy-4-methoxybenzaldehyde(**19**), exhibited $PLA_2$ inhibition (*in vitro*) and significantly reduced both lethality and hemorrhagic activity caused by *D. russelii* venom [69]. This compound was also extracted in substantial amounts from *Janakia arayalpatra* (Periplocaceae) in Jammu, India. This led researchers to synthesize semi-synthetic derivatives (**20-22**), which displayed venom-neutralizing activity against *D. russelii*. In *in vivo* experiments, these derivatives effectively counteracted hemorrhagic effects, lethality, and $PLA_2$-induced toxicity. Additionally, all of the synthesized compounds neutralized the hemorrhagic and lethal effects of *N. kaouthia* venom [70].

Another study revealed the venom-inhibitory properties of anisic acid (**25**) and salicylic acid (**26**), also derived from *H. indicus*. These compounds were found to counteract the venoms of *D. russelii*, *E. carinatus*, *N. kaouthia*, and *O. hannah*. Compound **23** completely neutralized the lethal effects and defibrinogenation induced by the venom in both *in vitro* and *in vivo* studies, while compound **24** entirely inhibited venom-induced hemorrhagic activity [66]. However, the precise mechanisms through which these compounds exert their venom-neutralizing effects remain unclear [66,71].

Gallic acid (**27**) was found to suppress the proteolytic activity of *D. russelii* venom in vitro ($IC_{50} = 0.58$ μM), though it did not suppress the $PLA_2$ activity of the same venom [72]. Nevertheless, a study by Pereañez et al., demonstrated that gallic acid inhibited $PLA_2$ enzymatic activity by 63% and reduced $PLA_2$-induced cytotoxicity by approximately 78% in *Crotalus durissus* venom [73]. Additionally, this compound, which was extracted from *Anacardium humile* (Anacardiaceae) in Minas Gerais, Brazil, exhibited strong myotoxic inhibition against the venom of *Bothrops jararacussu* and its principal myotoxins,

bothropstoxin I and II (BthTX-I and BthTX-II). Furthermore, gallic acid significantly reduced hemorrhagic activity ($IC_{50}=0.2$ μM) and oedema formation ($IC_{50}=2$ μM) in *in vivo* experiments [73].

Vanillic acid (**28**), on the other hand, displayed selective and specific inhibition of the enzymatic activity of 5'-nucleotidase (5'AMP), a venom enzyme known to affect hemostasis by preventing platelet aggregation. Found in the venom of *Naja naja*, this enzyme contributes to anticoagulant effects. In a dose-dependent manner, vanillic acid reduced the anticoagulant effect of *N. naja* venom by up to 40%. Research indicates that compound **28** interacts with one or more intrinsic blood clotting factors, disrupting the anticoagulant properties of the venom [74].

### 3.2.3. Hydroxycinnamic acids.

Rosmarinic acid (**29**), an esterified hydroxycinnamic acid found in various plants, is well known for its antivenom properties. When extracted from *Cordia verbenacea* (Boraginaceae), this compound was shown to inhibit both oedema and myotoxic activity caused by the basic $PLA_2$s, BthTX-I and BthTX-II. It was particularly effective in almost completely neutralizing the myotoxic effects and partially reducing edema triggered by these enzymes, suggesting a dissociation between their catalytic and pharmacological domains [75]. Additionally, rosmarinic acid has been effective against the venoms of *Agkistrodon bilineatus*, *Deinagkistrodon acutus*, *Crotalus atrox*, *Gloydius blomhoffii*, *Bitis arietans*, and *Trimeresurus flavoviridis* [76–78] (S1 Table and S1 Fig).

Caffeic acid (**30**) has also demonstrated venom-neutralizing potential. When complexed with piratoxin-I, a $PLA_2$ containing a lysine residue at position 49 ($PLA_2$s-Lys49) in *Bothrops pirajai* venom, it led to partial inhibition of PrTX-I's myotoxic activity [61]. Furthermore, this compound suppressed 41% of the cytotoxic activity induced by $PLA_2$s from *Crotalus durissus* [73].

A modified derivative, triacontyl *p*-coumarate (**31**), isolated from *Bombacopsis glabra* (Bombacaceae) in Bahia, Brazil, showed promising effects against the toxicity of *Bothrops pauloensis* venom, as well as isolated SVMPs (jararhagin) and $PLA_2$s (BnSp-6) [79]. Compound **31** significantly reduced fibrinogenolytic activity and plasma fibrinogen depletion (53%) induced by the venom or its isolated toxins. It also effectively inhibited hemorrhagic activity (3 MDH) and hemorrhagic effects caused by jararhagin [79] (S1 Table and S1 Fig).

The compound *p*-coumaric acid (**32**) formed a complex with $PLA_2$s from *Daboia russelii* and demonstrated strong catalytic inhibition, with an $IC_{50}$ value of 38.0 μM [80]. Chlorogenic acid (**33**) and cynarin (**24**) were also tested for their ability to inhibit venom-induced mortality in mice, with chlorogenic acid showing a high inhibition rate (90%) against *Bothrops jararaca* venom, while cynarin displayed a weaker effect (30%) [81].

Ferulic acid (**34**) exhibited significant $PLA_2$ inhibitory activity and effectively reduced oedema formation caused by $PLA_2$s from *Crotalus durissus* [82]. Another study reported that this compound inhibited $PLA_2$ enzymatic activity by 17% and reduced $PLA_2$-induced cytotoxicity by 37% in *C. durissus* venom [73]. In the same study, propyl gallate (**35**) was also shown to inhibit the enzymatic (51%) and cytotoxic (94%) activity of $PLA_2$s from this species (S1 Table and S1 Fig).

### 3.2.4. Tannins.

Several research studies have highlighted the role of some tannins as bioactive compounds with potential antivenom properties. Gallotannin tannic acid (**36**), which occurs naturally in various plants but is also commercially available, effectively inhibited hyaluronidases (HAases) and hemorrhagic effects while reducing the lethal impact of *Crotalus adamanteus* venom *in vivo*. This inhibition significantly increased the survival time of test mice [83].

Ellagitannins extracted from the leaves of *Casearia sylvestris* (family Salicaceae), collected in São Paulo, Brazil, were investigated for their ability to neutralize snake venom. Specifically, two compounds ellagic acid (**37**) and 3'-O-methyl ellagic acid (**23**) were evaluated for their inhibitory effects on the venom and the Asp49 phospholipase $A_2$ (BthTX-II) enzyme from *Bothrops jararacussu*. The inhibition constant (Ki) values for enzymatic activity were approximately 3 nM for **36** and nM for **37**. Additionally, their IC50 values for reducing oedematogenic and myotoxic activities were 23.8 μM and 34 μM, respectively [84] (S1 Table and S1 Fig).

Casuarictin (**38**), another ellagitannin, was sourced from the leaves of *Laguncularia racemosa* (Combretaceae) in São Paulo, Brazil. This compound formed a stable protein complex with $PLA_2$s isolated from *Crotalus durissus* venom. Molecular interactions between casuarictin and $PLA_2$s effectively neutralized the native oedematogenic effects and prevented

toxin-induced myonecrosis when administered 10 minutes after PLA$_2$s injection. As a result, casuarictin shows promise as a potential anti-inflammatory agent for treating PLA$_2$s-induced oedema and myonecrosis [85]. Another study investigated the ethanolic extract of *Mangifera indica* (Anacardiaceae) seed grains, particularly its primary active component, 1,2,3,4,6-pentagalloyl glucopyranose (**39**). This compound exhibited inhibitory effects on the enzymatic activities of PLA$_2$s, HAases, and L-amino acid oxidase (LAAO) from *Calloselasma rhodostoma* and *Naja naja* venoms in *in vitro* assays [86] (S1 Table and S1 Fig).

**3.2.5. Coumarins.** The incubation of BnIV, a Lys49 PLA$_2$ from *Bothrops neuwiedi* venom, with coumarin umbelliferone (**40**), a compound abundant in *Citrus* species, prevented platelet clumping, oedema (IC$_{50}$ 0.2 μM), and myotoxicity caused by BnIV while also reducing its inflammatory effects. Compound **40** interacted with Asp and Lys residues in the catalytic site of PLA$_2$s, which are crucial for the toxin's activity [87]. Among coumarins, the dihydrofuranocoumarin (+)-alternamin (**41**), a newly identified compound isolated from the aerial components of *Murraya alternans* (Rutaceae) in Myanmar, inhibited venom-induced bleeding by *Trimeresurus flavoviridis* by 24% at a concentration of 0.0827 m4M compared to the control [88]. Additionally, bergapten (**42**), derived from *Dorstenia brasiliensis* (Moraceae) in Brazil (Rio de Janeiro state), showed a modest effect in reducing the lethality of *Bothrops jararaca* venom in mice, decreasing lethality by 20% [81] (S1 Table and S1 Fig).

**3.2.6. Flavonoids.** Flavonoids exhibit a wide range of biological activities, with anti-inflammatory and antioxidant properties being particularly notable. These characteristics make several flavonoids promising candidates as antivenom agents. In this context, two thrombin-like serine proteases (SP1 and SP2) from *Crotalus simus* venom were isolated and incubated *in vitro* with the flavanone aglycone hesperetin (**43**), commonly found in *Rutaceae* species. The results showed that hesperetin acts as a potent non-competitive inhibitor against SP1 and a mixed inhibitor against SP2. This naturally occurring flavone, which can be easily extracted from oranges, presents a cost-effective option for inhibiting the studied snake venom proteases [89]. Additionally, hesperetin **43** inhibits phospholipase A$_2$ (PLA$_2$) activity in *Crotalus atrox* venom [87]. Furthermore, this compound, obtained from orange peels, functioned as a reversible inhibitor of serine proteases isolated from *Bothrops jararaca* venom [90].

The flavanone pinostrobin (**44**), extracted from the leaves of *Renealmia alpinia* (Zingiberaceae) in Colombia, a plant traditionally used in folk medicine for snakebite treatment, was evaluated for its effectiveness against the venoms of *Crotalus durissus* and *Bothrops asper*. Pinostrobin exhibited an IC$_{50}$ of 1.76 μM against PLA$_2$ activity in *C. durissus* venom. In experiments where mice were injected with PLA$_2$s and treated with pinostrobin at concentrations of 0.4, 2.0, and 4.0 μM, myotoxic activity was inhibited by up to 87% [91]. This compound also demonstrated the ability to inhibit the proteolytic effects (22%) induced by *B. asper* venom and reduced hemolytic activity by 21%. The same study found that *R. alpinia* extract inhibited the indirect hemolytic, coagulant, and proteolytic activities of *B. asper* venom following *in vitro* pre-incubation.

The glycosylated flavanone hesperidin (**45**), isolated from *Citrus sinensis* (*Rutaceae*), exhibited moderate inhibitory activity against the lethality induced by *B. jararaca* venom [81]. Synthetic flavones, including the aglycones apigenin (**46**) and luteolin (**47**), inhibited the hyaluronidase and hemorrhagic actions of *Crotalus adamanteus* venom and reduced its lethality [83,92]. Additionally, synthetic apigenin (**46**) was found to inhibit PLA$_2$ activity in *C. atrox* venom [93]. The flavones pectolinarigenin (**48**) and hispidulin (**49**), extracted from *Aegiphila integrifolia* (Verbenaceae) in Brazil (Roraima state), partially inhibited PLA$_2$ activity in *Bothrops atrox* venom by 20% and 15%, respectively, while also reducing hyaluronidase activity by 60% and 40%, respectively [93].

Another flavone, morelloflavone (**50**), a dimeric flavone isolated from *Garcinia madruno* (Clusiaceae) in Colombia, exhibited IC$_{50}$ values of 0.48 μM and 0.38 μM in PLA$_2$ enzymatic activity for aggregate and monodisperse substrates, respectively [94]. Flavonols have been extensively studied due to their frequent occurrence and accessibility through phytochemical approaches. The aglycone quercetin (**51**), isolated from *Morus nigra* (Moraceae) [95], *Phyllanthus klotzschianus* (Phyllanthaceae) [81], and *Erythroxylum ovalifolium* (Erythroxylaceae) [96], demonstrated potent inhibition of

oedema, proteolytic activity, and lethality caused by the venoms of *Bothrops jararacussu*, *Lachesis muta*, and *B. jararaca*, respectively. Synthetic quercetin (**52**) also inhibited PLA$_2$ activity in *Daboia russelii* venom (IC$_{50}$ 2 µM) and *Naja naja* venom, achieving a maximum inhibition of 40% [97]. Additionally, research by Cotrim and colleagues highlighted quercetin's potential to inhibit PLA$_2$ activity in *D. russelii* venom and its ability to prevent platelet aggregation and myotoxic effects induced by *C. durissus* venom by approximately 40% [98]. Quercetin also completely inhibited purified hyaluronidase activity in *N. naja* venom [99,100].

Kaempferol (**53**) has demonstrated antivenom potential [83], as have the synthetic flavonoids fisetin (**54**) and myricetin (**55**), which exhibited inhibitory effects against PLA$_2$ activity in *C. atrox* venom. Additionally, myricetin **55** effectively inhibited the proteolytic and hemorrhagic activities of *B. atrox* venom, with IC$_{50}$ values of 150 µM and 1 µM, respectively [101]. The same study identified quercitrin (**56**), a synthetic glycosylated flavonol, as an effective inhibitor of *C. atrox* venom PLA$_2$s. Furthermore, quercitrin **56**, extracted from the leaves of *Brownea rosa-de-monte* (Fabaceae) in Panama, significantly inhibited the coagulant and hemorrhagic effects of *B. asper* venom. At a concentration of 0.1 µM, quercitrin extended plasma coagulation time by two to six times [102].

Another glycosylated flavonol, quercetin-3-O-rhamnoside (**57**), isolated from *Euphorbia hirta* (Euphorbiaceae) in India, showed strong inhibition (93%) of *N. naja* PLA$_2$ activity in vitro using egg yolk as a substrate and also reduced hyaluronidase and hemolytic activity [103]. Furthermore, quercetin-3-O-rhamnoside reduced edema and lethality, extending the survival of mice [103]. Rutin (**58**) exhibited a 28% inhibition of *L. muta* venom-induced hemorrhagic activity *in vivo* [96] and demonstrated 40% inhibitory effect against PLA$_2$ activity in *C. atrox* venom [97].

Taxifolin (**59**), a synthetic flavanonol, showed potential for inhibiting PLA$_2$ enzymatic activity in *C. atrox* venom [97]. Several studies have also identified flavan-3-ols as potential antivenom agents. Catechin (**61**), extracted from the stem of *Scolopia chinensis* (Salicaceae) in China, demonstrated inhibitory activity (16%) against snake venom phosphodiesterase I (PDE I) [104]. This compound also showed PLA2 inhibition in *C. atrox* venom and has been synthesized [97]. The flavan-3-ol gallocatechin (**62**), obtained from the leaf extract of *Schizolobium parahyba* (Fabaceae) in Brazil (Minas Gerais state), neutralized the biological and enzymatic effects of venoms and toxins from *B. jararacussu* and *B. neuwiedi* [105]. Gallocatechin exhibited significant inhibition of hemorrhagic and fibrinogenolytic activity of isolated SVMPs (Bjussu-MP-I, Bjussu-MP-II) and also reduced myotoxic effects induced by *Bothrops alternatus* venom and BnSP-6 (a Lys49 PLA$_2$ from *B. neuwiedi*) [105].

Epigallocatechin gallate (**63**), a major compound in *Ilex paraguariensis* (yerba mate), reduced the cytotoxic effects of myotoxic PLA$_2$s from *C. durissus* venom, achieving an IC$_{50}$ of 0.38 µM, indicating its potential as an antimyotoxic agent [73]. The chalcone butein (**64**), isolated from *Butea monosperma* (Fabaceae), inhibited the activity of daboxin P, a PLA$_2$, with an IC$_{50}$ of 541 µM. Moreover, it completely inhibited PLA$_2$ activity in *N. naja* venom (100%) and reduced activity in *Bungarus caeruleus* (49%), *D. russelii* (72%), and *Echis carinatus* (47%) at 1,200 µM [106]. Primetin(**65**), a flavonoid isolated from *Primula denticulata* (Primulaceae), was found to inhibit toxins from snake venom [62]. Vale et al., investigated the protective effects of Myricetin-3-O-Glucoside (**66**) from the leaf extract of *Schizolobium parahyba* (*Leguminosae*) against snake venoms [105]. The flavonoids hypolaetin-8-glucoside (**67**) was reported to have phospholipase A$_2$ activities. Hypolaetin-8-glucoside was isolated from *Sideritis mugronensis* (Lamiaceae) [107]. Isoquercitrin (**68**), a flavonoid extracted from the root of *Solanum incanum*, demonstrated inhibitory effects on acetylcholine responses. Additionally, it significantly reduced intestinal motility in mice, suggesting a potential neuromodulatory role in venom-induced physiological disturbances [108]. A structurally complex flavonoid, patuletin-3-O-α-L-rhamnopyranosyl-7-O-α-L-rhamnopyranoside (**69**), was identified in *Bryophyllum pinnatum* leaves and exhibited notable antivenom activity. It demonstrated potential by modulating venom-induced oxidative stress and enzymatic activity [109].

Iridin(**70**), a flavonoid, demonstrates the ability to bind to biological polymers, inhibiting PLA$_2$ activity. Flavonoids like iridin, along with other members of the class such as quercetin and luteolin, show potential as anti-snake venom agents due to their structural features that allow for effective interaction with venom components [110]. Quercetin-3-O-sophoroside

(**71**), isolated from ethanolic extracts of *Cissampelos pareira* aerial parts, demonstrated protective effects against proteolytic activity of *Bothrops diporus* venom [10]. Quercetin derivatives are known to inhibit lipoxygenase, and although specific $PLA_2$ inhibition was not confirmed for this glycoside, its structural features suggest potential to modulate venom-induced inflammation. Nishijima and colleagues identified amenthoflavone (**72**) as active principles in *Byrsonima crassa* (Malpighiaceae) against *N. naja* venom [111]. Myricetin (**73**) was potent against the hemorrhagic effects of *Bothrops jararaca* [111]. Isoquercitrin (**74**) from *Schizolobium parahyba* (leaves) demonstrated anti-hemorrhagic and anti-fibrinogenolytic activities against *Bothrops* venoms, including *B. jararacussu* and *B. neuwiedi*, at concentrations of 100–1250 µM [112]. Patuletin-3-O-Glycoside (**75**), isolated from the leaves of *Bryophyllum pinnatum*, was characterized as a bioactive compound with antivenom properties. Using HPLC-DAD-MS, researchers confirmed its presence and potential role in counteracting venom-induced toxicity. Its mechanism of action is likely related to flavonoid-mediated enzyme inhibition and antioxidant effects, contributing to venom neutralization [109] (S1 Table and S1 Fig).

**3.2.7. Isoflavonoids.** The isoflavone harpalycin 2 (**76**), extracted from the leaves of *Harpalyce brasiliana* (Fabaceae) in Ceará, Brazil, has been traditionally used in folk medicine as an anti-inflammatory remedy for snakebites. Research on its effects revealed promising biological activity. Compound **76** demonstrated inhibitory effects on the enzymatic activity, oedema formation, and myotoxic effects of phospholipase $A_2$s ($PLA_2$s) from *Bothrops pirajai*, *Crotalus durissus*, and *Naja naja* venoms. Specifically, harpalycin 2 (Har2) inhibited Piratoxin 3 (PrTX-III) from *B. pirajai* by 59%, *C. durissus* venom by 79%, and *N. naja* venom by 88%. Additionally, oedema induced in mouse paws by $PLA_2$s was significantly reduced in the early phase upon administration of Har2. The compound also mitigated the myotoxic effects of these $PLA_2$s [113,114].

Another isoflavone, 7,8,3'-trihydroxy-4'-methoxyisoflavone (**77**), isolated from *Dipteryx alata* (Fabaceae) in Tocantins, Brazil, demonstrated the ability to counteract neurotoxicity and myotoxicity in *B. jararacussu* venom, as observed in mouse phrenic nerve-diaphragm experiments [115]. When venom was pre-incubated with **77** (0.0493 mM), it reduced neuromuscular blockade by 84%. It also attenuated the effects of BthTX-I, the primary myotoxic $PLA_2$s in this venom. Histological evaluation of the diaphragm muscle treated with **77** showed that 91% of muscle fibers remained intact, compared to 50% damage in venom-exposed samples [115].

Beyond isoflavonoids, derivatives such as pterocarpans and coumestans have also demonstrated antivenom properties. Edunol (**78**), a pterocarpan isolated from the roots of *Brongniartia podalyrioides* (Fabaceae) in Mexico, decreased the lethality of *B. atrox* venom by 30% in mice that had received an $LD_{50}$ dose of venom and were subsequently treated with 3.1 mg/kg of the compound [116]. This substance was also found in the roots of *Harpalyce brasiliana* in Brazil and was later synthesized to yield greater quantities for further testing. Synthetic edunol exhibited significant antimyotoxic, antiproteolytic, and anti-$PLA_2$s properties [117]. The potency of these effects was further enhanced by modifying edunol into its bioisostere derivative (**79**), where the prenyl group was replaced by a benzyl group. Pre-incubation of venom with **79** *in vitro* resulted in complete inhibition of myotoxic activity, with an $IC_{50}$ of 9.97 µM. At a concentration of 100 mM, this pterocarpan also inhibited 65% of the phospholipase activity of *B. jararacussu* venom and over 80% of its proteolytic activity [117].

Additional prenylated pterocarpans, particularly those with modifications in ring A, have shown strong activity against *B. atrox* venom. Notably, cabenegrins A-I (**80**) and A-II (**81**), isolated from *Annona crassiflora* (Annonaceae), a medicinal plant widely used in northeastern Brazil for treating snakebites, emerged as promising lead compounds [118]. The coumestan wedelolactone (**82**), identified in *Eclipta alba* and *E. prostrata* (Asteraceae) from São Paulo, Brazil, has also demonstrated antivenom activity [95,119,120]. A study by Cedro et al., found that it effectively inhibited the myotoxic activity of $PLA_2$s from *C. durissus* and *B. jararacussu* (BthTX-I and II) [121]. Additionally, compound **82** significantly reduced the proteolytic, $PLA_2$s, and myotoxic activities of *B. jararacussu* venom, with an $IC_{50}$ of 1 µM [122].

An analogue of wedelolactone (**83**), synthesized with modified oxygenation patterns in rings A and D, exhibited activity at 30 µM. This compound effectively antagonized the release of creatine kinase (CK) induced by *B. jararacussu* venom in skeletal muscle. Its inhibitory effect on venom-induced myotoxicity ($IC_{50}$ = 1 µM) was comparable to that of wedelolactone

[122]. Moreover, **83** showed a lower affinity for benzodiazepine receptors, suggesting a reduced likelihood of central nervous system side effects [122].These findings indicate that such compounds could be valuable in snakebite therapy and the treatment of coagulation disorders [122]. Furthermore, another study confirmed the inhibitory potential of demethylwedelolactone (**84**), also derived from *E. alba*, in neutralizing myotoxicity induced by isolated PLA$_2$s (BthTX-I and II) from *C. durissus* and *B. jararacussu* venom [120]. Pterocarpan (**85**), exhibited promising antivenom potential. Its pharmacological effects included modulation of inflammatory responses, inhibition of venom enzymes, and neutralization of oxidative stress induced by envenomation. These findings highlight its significance in the search for plant-based antivenom agents [64] (S1 Table and S1 Fig).

**3.2.8. Modified glycosides.** The compound 2-(6-benzoyl-β-glucopyranosyloxy)-7-(1α,2α,6α-trihydroxy-5-oxocyclohex-3-enoyl)-5-hydroxybenzyl alcohol (**86**) was isolated from the bark and branches of *Bennettiodendron leprosipes* and *Flacourtia ramontchi* (Salicaceae), both traditionally used in folk medicine. This compound exhibited 14% inhibition against PDE-I [123]. In the same study, homaloside D (**87**) displayed a similar effect, inhibiting PDE-I by 13%. Additionally, itoside B (**88**) and itoside F (**89**), extracted from *Itoa orientalis* (Salicaceae), demonstrated inhibition levels of 21% and 13%, respectively, against PDE-I [123].

Although these modified glycosides exhibited activity, their potency was lower compared to other natural product classes. Two new phenolic glycosides, scoloquinenoside C (**90**) and scoloposide C (**91**) were isolated from the stem of *Scolopia chinensis* (Salicaceae) in China. Phenolic glycosides, in general, have demonstrated inhibitory effects against PDE-I derived from snake venom [124]. Another study also confirmed PDE-I inhibition from snake venom, where two newly identified phenolic glycosides, benzoylsalreposide (**92**) and salireposide (**93**), were evaluated. Both compounds, isolated from *Symplocos racemosa* (Symplocaceae) in Pakistan, exhibited an IC$_{50}$ value of 171 μM [122,125] (S1 Table and S1 Fig).

**3.2.9. Polyketides.** The natural naphthoquinone lapachol (**94**) was extracted from *Tabebuia impetiginosa* (Bignoniaceae) in Brazil (state of Rio de Janeiro) and has served as a precursor for developing new bioactive quinones [126]. In the same study, an analogue of this compound, **95**, exhibited the ability to counteract the proteolytic activity (3–100 μM) and collagenase activity (10–100 μM) of *Bothrops atrox* venom. Furthermore, *in vivo* pre-incubation of the venom with **95** at doses of 1 mg/kg and 3 mg/kg prevented hemorrhage caused by *B. atrox* venom. Against *B. jararaca* venom, the inhibition exceeded 70% when treated with 10 mg/kg of the compound. The authors suggested that **95**'s protective effect on the skin results from its ability to inhibit proteolytic and collagenase activity, making it a potential candidate for preventing vascular degradation [126].

Another naphthoquinone, isohemigossypolone (**96**), was isolated from the roots of *Pachira aquatica* (Malvaceae) in Brazil (state of Bahia). This compound significantly reduced the *in vitro* coagulant activity of *Bothrops pauloensis* venom. *In vivo* tests further demonstrated its ability to inhibit the venom-induced myotoxic effects and neutralize 70% of the metalloproteinase activity of whole venom, while reducing the activity of the isolated BthMP enzyme by 40% [127].

Selvanayagam et al. identified a quinonoid xanthene from the root bark of *Ehretia buxifolia* (Boraginaceae) in India (Tamil Nadu), a plant traditionally used as an antidote for *Echis carinatus* envenomation [128]. The isolated compound, ehretianone (**97**), was obtained from the methanolic extract of the bark and tested for antivenom properties against *E. carinatus* venom in mice. In a prophylactic treatment, administering 3.75 mg/kg of the compound 30 minutes before venom injection reduced mortality by 35% compared to untreated controls. In a curative study, the same dosage provided significant protection when given within 5 minutes after venom injection [128].

Melanin (**98**), extracted from black tea, is a non-hydrolyzed complex of tea polyphenols that has been investigated for its effects on the venoms of *Agkistrodon contortrix*, *Agkistrodon halys*, and *Crotalus atrox*. *In vitro* assays showed that it reduced the specific activity of the PLA$_2$s enzyme by 43%, while in vivo experiments demonstrated a notable increase in the survival time of mice following administration of venom from these three snake species [129] (S1 Table and S1 Fig)

**3.2.10. Terpenoids.** Salama and colleagues isolated, labdane lactone (**99**), and labdane trialdehyde (**100**), from various *Curcuma* species (family Zingiberaceae) in Thailand. Compounds **99** and **100**, obtained from *C. antinaia* and *C. zedoaroides*, demonstrated venom inhibition rates of 83% and 62%, respectively, at a concentration of 22.7 µM and 21.9 µM against *Ophiophagus hannah* venom. Notably, compound **100** preserved diaphragmatic contraction almost entirely, offering 99% protection. This makes it the most promising candidate for antivenom and anti-neurotoxic therapy [130].

The sesquiterpene α-turmerone (**1001**), extracted from *C. longa* (Zingiberaceae) in Brazil, neutralized hemorrhagic activity in *Bothrops jararaca* venom and reduced *Crotalus durissus* venom lethality by 70% in rats [131]. It also decreased hemorrhage and oedema from *B. alternatus* venom, reversing necrosis in 96 hours [132]. The diterpene (E)-17-ethyliden-labd-12-ene-15,16-dial (**102**), from *C. zedoaroides* (Zingiberaceae) in Thailand, fully neutralized *Ophiophagus hannah* venom *in vitro* at 0.0704 mM and increased survival by >80% in mice [133]. It guided dialdehyde to the venom's peptide target, preventing neurotoxic effects. Further studies on diterpenes **103–105** (from *Curcuma* species) showed significant antivenom activity, with compound **105** offering the highest protection (99%).

The neo-clerodane (**106**) from *Baccharis trimera* (Asteraceae) in Brazil inhibited hemorrhagic and proteolytic activities in *Bothrops* venoms, with over 80% inhibition of SVMPs [133]. Among triterpenes, lupeol acetate (**107**) from *Hemidesmus indicus* neutralized lethal, hemorrhagic, and neurotoxic effects of *Daboia russelii* and *Naja kaouthia* venoms [134]. Compounds **108–109** from *Clematis gouriana* (Ranunculaceae) in India effectively inhibited PLA$_2$s. Oleanolic acid (**110**) from medicinal plants inhibited PLA$_2$s in *D. russelii* and *N. naja* venoms (>90%), reducing venom-induced oedema and hemorrhagic effects [130].

Betulinic (**111**) and ursolic acids (**112**) inhibited proteolytic and hemorrhagic activities in *Bothrops atrox* venom, showing strong antivenom properties [101]. Quinovic acid (**113**) (from *Mitragyna stipulosa* (Rubiaceae) inhibited venom PDE-I [135]. Arjunolic acid (**114**) from *Combretum leprosum* (Combretaceae) in Brazil reduced lethality (>80%) and inhibited multiple enzymatic activities in *Bothrops* venoms [136]. Friedelin (**115**) and lupeol (**116**) from *Erythroxylum* species in Brazil showed modest antivenom activity [96]. Triterpenoids from *D. alata* (Fabaceae) countered *B. jararacussu* neuromuscular blockade, with compounds **117–118** offering 45–70% protection. β-amyrin (**119**) from *Apuleia leiocarpa* (Fabaceae) improved survival against *B. jararaca* venom [95].

Among steroids, ikshusterol 3-O-glucoside (**120**) from *C. gouriana* neutralized *N. naja* venom moderately. β-sitosterol (**121**) and stigmasterol (**122**) from *Pluchea indica* protected against *D. russelii* and *N. kaouthia* venom effects. The synthetic corticosteroid corticosterone (**123**) inhibited PLA$_2$s in *D. russelii* venom. Bakuchiol (**124**), a synthetic meroterpenoid, inhibited the main PLA$_2$ enzyme *daboxin P* [80,106,137]. Bt-CD (7α-hydroxy-3,13-clerodadiene-16,15:18,19 diolide) (**125**), a clerodane diterpenoid isolated from *Baccharis trimera*, displayed anti-proteolytic and anti-hemorrhagic properties against Bothrops venom. It effectively neutralized hemorrhagic and fibrinogenolytic activities of P-I and P-III metalloproteinases from *Bothrops neuwiedi* and *Bothrops jararacussu* venoms [62]. The furanoid diterpene lactone (**126**) from *Aristolochia albida* (rhizome) provided *in vivo* protection from venom-induced mortality caused by *Naja nigricollis* and *Bitis arietans*, with an effective dosage range of 5–80 mg/kg [112]. 2-Hexenal (**127**), 2-Hexen-1-ol (**128**), Elixene (**129**), (−)-Globulol (**130**), and Thujopsene (**131**) isolated from *Bidens pilosa* L. leaves and whole plant extracts exhibited significant inhibitory activity against venoms from *Dendroaspis jamesoni* and *Echis ocellatus* [108].

d-α-pinene (**132**) and camphene (**133**), isolated from *Citrus limon* root and ripe fruit methanolic extracts, exhibited *in vitro* neutralization of *N. naja karachiensis* venom-induced coagulant effects, achieving 64% inhibition [10]. Furthermore, an ED$_{50}$ of 710 µg/mouse was reported against the lethal effects of *Lachesis muta* venom. These compounds may disrupt enzymatic venom components or enhance host defense mechanisms via membrane stabilization. d-Limonene (**134**), isolated from the methanolic extracts of *Citrus limon* L. Burm. f. (roots and ripe fruits), demonstrated promising antivenom activity. It effectively neutralized the coagulant effect of weak phospholipase A$_2$ (PLA$_2$) enzymes found in *Naja naja karachiensis* venom, achieving 64% inhibition with a coagulation time of 109 ± 1.00 seconds. Furthermore, it exhibited

*in vitro* inhibitory effects against the lethal action of *Lachesis muta* venom, with an $ED_{50}$ of 710 µg of extract per mouse, indicating its potential utility as a broad-spectrum antivenom compound [10]. (S1 Table and S1 Fig)

**3.2.11. Saponins.** Certain saponins have also been identified as inhibitors of snake venom. The triterpene saponin bredemeyeroside B (**135**), derived from the roots of *Bredemeyera floribunda* (Polygalaceae) found in the Brazilian state of Ceará, demonstrated inhibitory effects against the lethal action of *Bothrops jararaca* venom when administered orally at a dose of 100 mg/kg. This resulted in a 90% survival rate among the tested animals [138]. Another compound from the same plant species, bredemeyeroside D (**136**), also exhibited venom lethality inhibition, with a 100% survival rate of mice after six hours under the same experimental conditions as the previous study [139].

Glycyrrhizin (**137**), obtained from the roots of *Glycyrrhiza glabra* (Fabaceae) in São Paulo, Brazil, demonstrated *in vitro* inhibition of human fibrinogen coagulation triggered by *B. jararaca* venom, with an $IC_{50}$ of 1.2 mM. Additionally, it showed hydrolytic activity *in vitro* ($IC_{50}$ 0.47 mM) and inhibitory effects on platelet aggregation ($IC_{50}$ 0.33 mM) [138]. In *in vivo* tests, compound **138** significantly reduced thrombus weight by 86% against venom-induced effects and successfully prevented venom-related bleeding when co-administered with antivenom [138]. Macrolobin A (**139**) and macrolobin B (**140**), isolated from the bark of *Pentaclethra macroloba* (Fabaceae) in Amapá, Brazil, effectively inhibited the hemorrhagic and fibrogenolytic activities of *Bothrops* venoms and the SVMP Bjussu-MP-I from *B. jararacussu* venom in a dose-dependent manner [140]. Compound **139** exhibited the most promising results, with inhibition of up to 90% of the raw venom's proteolytic activity and 80% of fibrin SVMPs, particularly classes I and III. In terms of coagulation activity, compound **139** completely inhibited the effects of *B. jararacussu* venom and the thrombin-like enzyme Bjussu-SP-I after incubation for one hour and 30 minutes, respectively [140].

Glycosidic derivatives of quinovic acid, initially isolated from the bark of *Bridelia ndellensis* (Euphorbiaceae) in Ngaoundéré, Cameroon, displayed inhibitory activity against phosphodiesterase-I (PDE-I). The compounds quinovic acid-3-O-α-L-rhamnopyranoside (**141**), quinovic acid-3-O-β-D-fucopyranoside (**142**), and quinovic acid-3-O-β-D-glucopyranosyl (1→ 4)-β-D-fucopyranoside (**143**) significantly inhibited the enzyme, with $IC_{50}$ values of 85 µM, 85 µM, and 75 µM, respectively [141]. A triterpenoid saponin, the potassium salt of gymnemic acid (**144**), was isolated from *Gymnema sylvestre*. This compound inhibited ATPase activity induced by *Naja naja* venom [142] (S1 Table and S1 Fig).

**3.2.12. Other compounds.** Fozing et al., demonstrated that certain compounds extracted from the leaves of *Morus mesozygia* (Moraceae) inhibit the phosphodiesterase I enzyme of commercial origin. Among them, mesozygin B (**145**) and artonine I (**146**) exhibited the strongest activity, with $IC_{50}$ values of 8.9 µM and 15.4 µM, respectively [143]. Other compounds have also been shown to effectively inhibit enzymes found in snake venom. The stilbene resveratrol (**60**) and the aliphatic derivative polyamine gramine {3-[3-(dimethylaminomethyl)-1H-indol-7-YL] propan-1-ol} (**147**) were tested for their ability to inhibit the $PLA_2$s enzyme from *Daboia russelii* venom, with $IC_{50}$ values of 43.4 µM and 50.4 µM, respectively [80].

In *in vitro* assays, the amino acid mimosine [β-3-(3-hydroxy-4-oxopyridyl) α-amino-propionic acid] (**148**) inhibited HAases (DRHyal-II) in a dose-dependent manner, achieving complete inhibition at 24 µM and an $IC_{50}$ of 12 µM. Additionally, compound **148** neutralized the hyaluronidase activity of *D. russelii* venom, which was entirely suppressed at 1000 µM, with an $IC_{50}$ of 500 µM. In *in vivo* experiments, **148** also reduced the myotoxic effects of the myotoxin VRV-PL-VIII (a myotoxic $PLA_2$s), which was potentiated by DRHyal-II [144]. According to a study by Devi et al., this synthetic compound inhibited the $PLA_2$s enzyme activity of raw venom (0.0173 mM) from *Naja naja* (47.0%), *Echis carinatus* (47.0%), and *D. russelii* (27%) at a concentration of 1,200 µM. Furthermore, compound **148** inhibited the $PLA_2$s activity of daboxin P with an $IC_{50}$ of 674.3 µM, indicating its anticoagulant potential [106].

Another study confirmed the medicinal value of diisobutyl phthalate, 2-methylpropyl phthalate, isolated from the root of *Emblica officinalis* (Phyllanthaceae) in India, which counteracted the myotoxicity induced by *D. russelii* venom. This was evidenced by a reduction in the levels of myotoxicity marker enzymes CPK and LDH [145]. Similarly, the bioactive polyphenol curcumin (**149**) exhibited strong inhibition of hyaluronidase (HAases) activity purified from *N. naja* venom, showing a 91% reduction [100].

Additionally, plant extracts commonly used in traditional medicine were tested for their ability to inhibit enzymatic activity in myotoxin I and PLA$_2$s from *Bothrops asper* venom. The compound 4-nerolidylcatechol (**150**), obtained from *Piper umbellatum* and *P. peltatum* (Piperaceae) in Costa Rica (Upala and Guápiles), inhibited the PLA$_2$s activity of myotoxin I from *B. asper* and *B. atrox* in a time- and dose-dependent manner (IC$_{50}$ = 1 mM). This compound was also effective against PLA$_2$s activity in group I pseudexin and *Micrurus mipartitus* venom, as well as group II *Bothrops* toxins [146]. Pre-incubation of **151** with myotoxins from venoms of these snake species reduced myotoxic activity by approximately 50% and significantly decreased the inflammatory response in both cases [146]. Additionally, in *in vivo* studies, pre-treatment with 0.4 mg of **151** up to one hour before toxin administration resulted in a one-third reduction in venom-induced oedema. Research by Nuñez et al., also demonstrated that this compound inhibited the proteolytic activity of trypsin on casein by 75% and eliminated the procoagulant action of an SVSP enzyme isolated from *B. jararacussu* venom in *in vitro* tests [146] (S1 Table and S1 Fig).

The fatty alcohol 1-hydroxytetratriacontan-4-one, extracted from the leaves of *Leucas aspera* (Lamiaceae), exhibited strong venom-neutralizing effects against *N. naja* in *in vivo* tests. It had a median effective dose (ED$_{50}$) of 34.47 mg and a fully effective dose (ED$_{100}$) of 68.93 mg per animal. A 100% survival rate was achieved when envenomed mice were treated simultaneously with **152** at a dose of 75 mg per animal. Additionally, it significantly reduced venom-induced oxidative stress and lipid peroxidase activity in multiple organs [60]. 12-Methoxy-4-Methylvoachalotine (MMV) (**152**) a natural product from *Tabernaemontana catharinensis*, has demonstrated inhibition of myotoxic PLA$_2$ activity and reduction in the lethality of *Crotalus durissus terrificus* venom. Its activity further supports the role of alkaloid-rich plant extracts in mitigating venom-induced toxicities [65].

Silybin (I) (**153**), isolated from *Silybum marianum*, has been reported for its potential anti-venom properties. However, further investigations are needed to establish its mechanism of action in venom neutralization [65]. Phloretin (**154**), classified as a dihydrochalcone, has been identified with inhibitory activity against SVHYs (snake venom hyaluronidases).

*n*-Propyl Gallate (**155**), a known antioxidant, exhibited inhibitory potential against *Naja naja* venom Haase [65]. Butylated Hydroxytoluene (BHT) (**156**), a synthetic antioxidant, has been studied for its potential to inhibit venom HAases. While it demonstrated some inhibitory activity, its efficacy was lower than that of NDGA and curcumin, suggesting a limited role in snakebite treatment [65]. Methyl gallate (**157**), isolated from *Mangifera indica* seed kernel extract, exhibited inhibitory effects against the proteolytic activity of *Calloselasma rhodostoma* and *Naja naja kaouthia* venoms. The presence of pentagalloylglucopyranose (tannic acid) in this extract significantly enhanced its activity, suggesting synergistic effects in venom neutralization [62]. Raghavamma and colleagues reported the antivenom activiy of α-amyrin (**158**) from the leaf extract of *Pergularia daemia* (Forsk.) Chiov [147].

Sarkhel et al., isolated di-iso-butyl phthalate (**159**) from the root extract of *Emblica officinalis* (*Phyllanthus emblica*), a species in the Euphorbiaceae family. This compound demonstrated activities against viper and cobra venoms, neutralizing venom-induced lethality, coagulant activity, defibrinogenation, fibrinolysis, free radical generation, hemorrhage, and myotoxic effects [145].

Lattmann and colleagues isolated curcuma dialdehyde (**160**) from *Curcuma zedoaria*. This C20 dialdehyde exhibited significant *in vitro* and *in vivo* antivenom activity against *Ophiophagus hannah* (King cobra) venom. The compound antagonized the venom's effects on neuromuscular transmission in isolated rat phrenic nerve-hemidiaphragm preparations [133]. Short-chain hydrocarbons 2,4-dimethylhexane (**161**), 2,6-dimethylheptane (**162**), and 2-methylnonane (**163**) were isolated from *Tragia involucrata*. These compounds demonstrated PLA$_2$ neutralization activities in mice, reducing venom-induced toxicity [62]. Dharmappa and colleagues extracted genistein (**164**) from *Glycine max*. The compound exhibited strong inhibitory effects against *Naja naja* phospholipase-I and RV-PL-V, significantly reducing venom-induced paw edema in mice [148]. Mahadeswaraswamy et al., isolated mimosine, a β-3-(3-hydroxy-4-oxopyridyl) α-amino propionic acid (**165**), from *Mimosa pudica*. The compound exhibited significant *in vitro* inhibition of hyaluronidase activity induced by *Vipera russelli* venom [144]. Leucasin (**166**), extracted from *Leucas aspera* (Lamiaceae) leaves exhibited

activity against venom protease and hyaluronidase activities. Studies identified leucasin in *L. aspera* extract as the active component. Leucasin neutralized $PLA_2$ type I enzyme from snake venom and demonstrated a dose-dependent antioxidant activity [149]. AIPLAI (**167**), isolated from *Azadirachta indica*, exhibited anti-$PLA_2$ activity against *Naja kaouthia* and *Daboia russelii* [150]. Catechin-3'-O-rhamnopyranoside (**168**), found in the stem of *Neocarya macrophylla*, inhibited $PLA_2$ activity in *Naja nigricollis* venom at concentrations of 1–10 mg/kg [151].

Quinonoid xanthene ehretianone (**169**), isolated from *Ehretia buxifolia*, exhibited anti-snake venom activity. The $LD_{50}$ of the venom used was 6.65 mg/kg, and the compound did not show any toxic effects up to 100 mg/kg [112]. An organic acid (designated HI-RVIF) (**170**), isolated from the root extract of *Hemidesmus indicus* (Indian sarsaparilla), displayed inhibitory activity against viper venom-induced lethal, hemorrhagic, coagulant, and anticoagulant effects in rodents. Spectral analysis confirmed the presence of a benzene ring, methoxy group, and hydroxyl group. Alpha-tocopherol (**171**), the most active form of vitamin E, inhibited $PLA_2$ activity selectively toward lamellar fluid membranes, thereby protecting phospholipids from enzymatic degradation. This compound reduced both the initial rate and overall extent of hydrolysis. The inhibition was non-competitive, suggesting that alpha-tocopherol mainly affected the substrate (membrane) rather than the enzyme itself [152].

Cardiac Glycosides (**172**) isolated from the leaves of *Costus afer* demonstrated significant *in vivo* protection against snake venom-induced toxicity. This compound exhibited potential antivenom properties, through their ability to inhibit venom enzymes and modulate physiological responses to envenomation. Their efficacy suggests a promising avenue for developing plant-based treatments for snakebite management [153]. D-Mannitol (**173**), a naturally occurring sugar alcohol extracted from *Mimosa pudica*, was found to possess significant antivenom properties. It functioned by counteracting the toxic effects of venom components, potentially through osmoregulatory mechanisms and interaction with venom enzymes. Its bioactivity suggests that it may serve as a complementary agent in snakebite treatment strategies [153].

Polyisoprenylated Benzophenones (**174**) isolated from the leaves, stems, flowers, and fruits of *Clusia fluminensis* exhibited *in vivo* venom-neutralizing activity against *Bothrops jararaca* venom. This compound demonstrated potential by inhibiting enzymatic pathways involved in venom toxicity, thereby reducing the systemic damage caused by envenomation. Their pharmacological potential underscores their significance in natural antivenom research [153].

2-Hydroxy-4-Methoxy Benzoic Acid (**175**), isolated from *Hemidesmus indicus*, exhibited strong bioactive properties that contributed to its antivenom potential. This phenolic compound likely exerts its protective effects through enzyme inhibition and antioxidant mechanisms, reducing venom-induced tissue damage and systemic toxicity [64]. D-X-Pinene Camphene (**176**) extracted from *Citrus* spp. (*C. limon L. Burm. F*) roots and ripe fruits using methanol, this compound effectively neutralized the anticoagulant effect induced by weak $PLA_2$ enzymes from *Naja naja karachiensis* venom, achieving 64% inhibition with a coagulation time of 109 ± 1.00 s. Additionally, it demonstrated *in vitro* inhibition of the lethal effects of *Lachesis muta* venom, with an effective dose ($ED_{50}$) of 710 µg extract per mouse [108].

Ichangin 4-β-Glucopyranoside (**177**) derived from *Citrus* spp. (*C. limon L. Burm. F*) root and ripe fruit extracts using methanol, this compound exhibited anticoagulant neutralization against weak $PLA_2$ enzymes in *Naja naja karachiensis* venom (64% inhibition, coagulation time of 109 ± 1.00 s). It also displayed inhibitory potential against *Lachesis muta* venom lethality, with an effective dose of 710 µg extract per mouse [108].

Nomilinic Acid (**179**) isolated from *Citrus* spp. (*C. limon L. Burm. F*) root and ripe fruit extracts in methanol, nomilinic acid significantly neutralized anticoagulant effects caused by weak $PLA_2$ enzymes in *Naja naja karachiensis* venom (64% inhibition, coagulation time of 109 ± 1.00 s) and demonstrated *in vitro* inhibition of *Lachesis muta* venom lethality at an effective dose of 710 µg extract per mouse [108].

Lignan (-)-Cubebin (**180**) derived from *Aristolochia* spp. (*A. indica*, *A. odoratissima*) leaf extracts using methanol, ethanol, pentane, and water, (-)-cubebin inhibited $PLA_2$ and hyaluronidase enzymes from *Naja naja* and *Vipera russelli* venoms. It also exhibited strong gelatinolytic, collagenase, peroxidase, and nuclease inhibitory activities and protected mice from lethal effects of *Bothrops atrox* venom at higher doses of 8 and 16 mg/kg [108]. 2-Mercapto-L-Cysteine (**181**)

extracted from *Allium sativum* L. bulbs using methanol. The compound provided 50% protection against *Naja naja karachiensis* PLA$_2$, evidenced by an increase in pH of an egg yolk suspension. It also neutralized the anticoagulant effect induced by weak phospholipase A enzymes in *Naja naja karachiensis* venom, achieving 40% inhibition with a coagulation time of 115 ± 1.52 s [108].

Sativin I (**182**) extracted from *Allium sativum* L. bulbs using methanol, Sativin I exhibited hepatoprotective activity, as assessed through AST and ALT enzyme levels. It provided 50% protection against *Naja naja karachiensis* PLA$_2$ by increasing the pH of an egg yolk suspension and neutralizing its anticoagulant effects (40% inhibition, coagulation time of 115 ± 1.52 s) [108]. Sativin II (**183**) derived from *Allium sativum* L. bulbs using methanol, Sativin II displayed hepatoprotective properties, as observed in AST and ALT enzymatic evaluations. It offered 50% protection against *Naja naja karachiensis* PLA$_2$, resulting in increased pH of an egg yolk suspension and a 40% inhibition of the anticoagulant effect, with a coagulation time of 115 ± 1.52 s [108].

8-Methoxycoumestrol (**184**), present in low concentrations in *Medicago sativa* L., was prepared as a sodium salt derivative. It mimicked wedelolactone by exhibiting antimyotoxic activity and inhibiting edema, hemorrhagic effects, and cardiotoxicity induced by *Bothrops jararacussu* crude venom.[154] 1-Hydroxytetratriacontane-4-One (**185**) isolated from *Leucas aspera* Linn. methanolic leaf extract, this compound significantly neutralized the lethal activity of *Naja naja naja* venom in a mouse model [154]. 3,4-Dihydroxyphenyllactic Acid (**186**), a derivative of rosmarinic acid, was first reported in *Cordia verbenacea*. It exhibited anti-inflammatory and antimyotoxic properties against *Bothrops jararacussu* venom. It also inhibited the myotoxic and oedema-inducing activities of Lys49 PLA2 and Asp49 isoforms from *B. jararacussu* venom [154].

Caftaric Acid (**187**) exhibited neuromuscular blocking activity and inhibition of muscle damage induced by Lys49-PLA$_2$ (*BthTX-I*) from *Bothrops jararacussu* venom [155]. Chicoric Acid (**188**) has demonstrated significant *in vitro* inhibitory effects on neuromuscular block and muscle damage, as evidenced by structural and calorimetrical assays. It was analyzed in complex with Lys49-PLA$_2$ (BthTX-I) from *Bothrops jararacussu* venom through small angle X-ray scattering and dynamic light scattering, providing insights into the oligomeric behavior of the PLA$_2$ and acid complex. The compound shows promise as a potential therapeutic agent in mitigating the effects of snakebite toxins [155].

Triacontyl p-Coumarate (**189**) as a derivative of p-Coumaric acid, triacontyl p-Coumarate (PCT) demonstrated a 45% inhibition of the coagulant activity of *Bothrops pauloensis* venom in bovine plasma. PCT effectively prevented fibrinogen degradation *in vitro* by SVMPs such as Jararhagin and BleucMP, providing promising evidence of its potential in mitigating the harmful effects of snake venom, particularly hemorrhagic activity [155]. 2-Hydroxy-4-methoxybenzaldehyde (**190**), isolated from *Janakia arayalpathra*, a plant used as an antidote in Western India, exhibits anti-snake venom activity. It targets phospholipase A$_2$ (PLA$_2$), a major component of snake venom, and holds potential as a therapeutic agent in treating venom-induced pathophysiological changes. Its application could be valuable in developing new anti-snake venom treatments [70].

2-Hydroxy-3-methoxybenzylalcohol (**191**) was found to have significant anti-venom activity, including anti-hemorrhagic and PLA$_2$ inhibitory effects, in *in vivo* and *in vitro* models. The compound's activity highlights its potential in treating venom-induced pathophysiological changes. Its structure, featuring a formyl group and a phenolic group, suggests it could be a valuable lead in anti-snake venom drug development [70].

2-OH-4-MeO Benzaldehyde (**192**), also isolated from *Janakia arayalpathra*, displays inhibitory activity against PLA$_2$ enzymes found in snake venom. Its promising antivenom properties make it a candidate for further investigation as a therapeutic agent against venom-induced envenomations [70]. Coumestrol (**193**), a coumestan derived from *Eclipta prostrata*, exhibits anti-snake venom properties, particularly against the venom of *Bothrops jararacussu*. While less potent than wedelolactone, coumestrol's antagonistic effects on creatine kinase release induced by venom suggest it could serve as a potential therapeutic in snakebite treatment [110].

Iso Butyl Phthalate (**194**), although not detailed in its exact mechanism, has been identified as a component with antivenom activities in various medicinal plants. It contributes to the broader class of molecules that inhibit the toxic effects

of snake venoms [110]. Dimercaprol (**195**), part of a group of small-molecule snake toxin inhibitors, has shown potential in clinical settings, particularly when combined with other inhibitors like varespladib. The compound's efficacy in neutralizing venom toxins and its progression to Phase II clinical trials suggest that it may be a valuable therapeutic in snakebite management [110]. Manoalide (**196**), isolated from *Curcuma longa*, exhibits potent anti-cobra $PLA_2$ activity. Its therapeutic potential as a natural venom inhibitor positions it as a key component in developing treatments for snakebite envenomations, particularly in cases involving cobras [156].

C20 dialdehyde (**197**), isolated from *Curcuma zedoaroides*, has demonstrated *in vitro* and *in vivo* activity against King cobra venom. The compound effectively inhibits neuromuscular transmission, offering significant promise for the treatment of venom-induced paralysis and other physiological disruptions caused by snake venom [133]. Protocatechuic acid (**198**) was reported for its cardioprotective and antihemolytic activities in the context of *Naja naja karachiensis* venom exposure. Extracts containing this compound significantly stabilized red blood cell membranes and inhibited venom phospholipase $A_2$ ($PLA_2$) activity, resulting in 76% coagulation inhibition [10]. Its antioxidant properties are believed to underlie its capacity to neutralize oxidative and inflammatory components of venom, although cytotoxicity was noted at higher concentrations (>1.57 mM).

Ichangin-4-β-glucopyranoside (**199**), a glycoside compound identified in the methanolic extract of *C. limon* roots and ripe fruits, also demonstrated antivenom efficacy comparable to d-limonene. The compound achieved 64% inhibition of $PLA_2$-mediated coagulation from *N. naja karachiensis* venom, with a coagulation time of $109 \pm 1.00$ seconds. In parallel, it mitigated the lethal effect of *L. muta* venom in vitro, with an $ED_{50}$ of 710 µg per mouse. These findings support its role as a potential lead compound in venom neutralization strategies [10]. 7-α-H-Cyclopenta[a]cyclopropa[f]cycloundecene-2,4,7,7a,10,11-hexol (**200**), derived from aqueous root extracts of *Cynodon dactylon*, exhibited significant antihemolytic activity against crude *Naja naja* venom, ranging from 84.3% to 90.6% inhibition across concentrations of 200–1,500 µg/ml. Such potent membrane-stabilizing effects suggest its potential as a protective agent against venom-induced hemolysis [10]. α-D-glucopyranoside, O-α-D-glucopyranosyl-(1→3)-α-D-fructofuranosyl (**201**) a disaccharide compound, isolated from aqueous extracts of *C. dactylon* roots, demonstrated comparable antihemolytic activity (84.3%–90.6%) against *N. naja* venom. The sugar moiety's bioactivity may stem from its ability to chelate venom components or interfere with cell membrane interactions, thus reducing venom-induced erythrocyte lysis [10].

9-Octadecenoic Acid (2-phenyl-1,3-dioxolan-4-yl)methyl Ester (**202**), identified in *C. dactylon* aqueous root extract, this lipid-based ester showed robust antihemolytic action (84.3%–90.6%) against *N. naja* venom. The amphipathic nature of this molecule likely contributes to its membrane-protective activity by interacting with venom enzymes or lipid bilayers [10].

9,10-Secocholesta-5,7,10[19]-triene-1,3-diol,25-[(trimethylsilyl)oxy] (3α,5Z,7E) (**203**), derived from *C. dactylon* root extract, significantly protected against hemolysis caused by crude *N. naja* venom, with inhibition levels between 84.3% and 90.6%. Its triene structure and hydroxyl substitutions may facilitate effective neutralization of venom phospholipases or pore-forming toxins [10]. Octasiloxane, 1,1,3,3,5,5,7,7,9,9,11,11,13,13,15,15-hexadecamethyl (**204**) exhibited high antihemolytic potential (84.3%–90.6%) in response to *N. naja* venom exposure. Its hydrophobic silicon-based backbone may inhibit venom-induced hemolysis by disrupting protein–membrane interactions or stabilizing erythrocyte membranes [10]. Oximino-2,7-diethoxyfluorene (**205**) isolated from the aqueous extract of *C. dactylon* roots, this fluorene derivative significantly prevented hemolysis caused by *N. naja* venom *in vitro*. Its inhibition percentage (84.3%–90.6%) underscores its effectiveness, potentially due to interference with venom-induced oxidative or lytic pathways [10].

Hexadecanoic acid, 1-(hydroxymethyl)-1,2-ethanediyl ester (**206**), identified from aqueous root extracts of *Cynodon dactylon*, demonstrated significant antihemolytic activity, with inhibition rates ranging from 84.3% to 90.6% against *Naja naja* venom at concentrations between 200 and 1,500 µg/mL. The compound's potent membrane-stabilizing potential is presumed to underlie its hemolytic suppression, although the exact mechanism remains to be elucidated [10]. Hexasiloxane, 1,1,3,3,5,5,7,7,9,9,11,11-dodecamethyl (**207**) also derived from the aqueous extract of *Cynodon dactylon* roots, hexasiloxane exhibited similar antihemolytic efficacy (84.3%–90.6%) against crude *Naja naja* venom. As a siloxane-based

compound, its interaction with venom components may be attributed to its potential to interfere with lipid bilayers or protein conformation [10].

Heptasiloxane, 1,1,3,3,5,5,7,7,9,9,11,11,13,13-tetradecamethyl (**208**) isolated from *C. dactylon* aqueous root extract, heptasiloxane showed comparable antihemolytic activities (84.3% to 90.6%). These findings highlight the potential therapeutic value of siloxane derivatives as inhibitors of hemolysis, possibly via membrane-protective or surfactant-like behavior [10].

Estra-1,3,5 [10]-trien-17-α-ol (**209**) exhibited significant inhibition of venom-induced hemolysis when tested in the same concentration range as other compounds from *Cynodon dactylon*. Its structural similarity to hormones suggests possible interference with venom enzymes or oxidative pathways. Further pharmacological evaluations are required to determine its systemic safety and efficacy [10]. 3,3′,5′-Trimethoxy-4,5-methylenedioxy dihydrostilbene (**210**) isolated from the hexane fraction of aerial parts of *Indigofera capitata* (Fabaceae), this novel dihydrostilbene displayed *in vivo* antivenom activity against *Naja nigricollis* venom in murine models, with survival rates of 50% and 16.5% at 1 mg/kg and 5 mg/kg, respectively. The compound is a promising lead candidate for development of antivenom therapeutics due to its observed protective efficacy [157].

3,5-Dimethoxy-4-hydroxyphenylacetic acid (**211**) from *A. parvifolia*, this phenolic acid constituted 5.30% of the extract's content and likely contributes to the antioxidant and anti-PLA$_2$ effects observed. Phenolic compounds are known to inhibit venom enzymatic activity and reduce oxidative stress, enhancing their potential in venom neutralization [158]. Acetylmarinobufogenin (**212**) a dominant compound in the bark extract (14.89%), acetylmarinobufogenin showed strong inhibitory potential against PLA$_2$ venom enzymes with docking energies (−7.22 to −9.59 kcal/mol). Its steroidal structure may enhance lipophilic interaction with enzymatic active sites, aiding venom neutralization [158]. γ-Sitosterol (**213**) representing 10.44% of the extract's content, γ-sitosterol is a triterpenoid with known anti-inflammatory and membrane-stabilizing properties. While its specific antivenom mechanism is yet to be fully elucidated, its presence contributed to mitigating venom-induced pathophysiological changes [158] (S1 Table and S1 Fig).

### 3.3. Computational analysis of ADMET properties, medicinal chemistry, and drug-likeness

To evaluate and predict the ADMET profiles, drug-likeness, and medicinal chemistry properties of 213 plant-derived compounds tested against various snake venoms and toxins targets, the compounds were grouped based on their chemical classes. Table 1 provides a summary of the number of molecules that progressed through each stage and the final selection of compounds with strong pharmacokinetic and pharmacological potential.

**3.3.1. Drug-likeness evaluation of the compounds.** Drug-likeness assessments were initially performed using SwissADME, ADMETlab and DataWarrior. This analysis evaluated the molecules based on Lipinski, Veber, Ghose, Egan, and Muegge rules [17,159–161]. Molecules failing to meet the drug-likeness criteria specifically those violating more than one rule (Veber and Egan) or more than two rules (Lipinski, Ghose, and Muegge) were excluded from further consideration. Out of the 213 molecules classified into 11 chemical groups in this study, 97 passed the drug-likeness evaluation. The highest success rates were observed in the coumarin, benzenoids, polyketides (100%) and isoflavonoids (90%) classes, whereas the saponin (0%) class had the lowest number of passed compounds.

The high approval rate of coumarins, benzenoids, isoflavonoids, and polyketides can be attributed to their relatively small and stable structures, which align with key pharmaceutical viability criteria, enhancing their bioavailability and systemic distribution [162,163]. Moreover, the structural diversity within these classes enables chemical modifications that can further improve their pharmacokinetic properties [163]. In contrast, saponins exhibit large, complex molecular structures, often surpassing 500 Da in molecular weight, with high topological polar surface area (TPSA) values due to multiple hydrophilic functional groups [161,164]. Saponins also possess numerous hydrogen bond donors and acceptors, which significantly limit their permeability [163,165]. The findings from this initial stage are highly significant, as molecules that fail to meet drug-likeness criteria generally have a lower likelihood of becoming successful drug candidates and can

PLOS Neglected Tropical Diseases

**Table 1. Assessment of drug-likeness, ADMET characteristics, and medicinal chemistry parameters of plant-derived compounds with inhibitory potential against different snake venom/toxin targets.**

| Compound Class | DL | A | D | M | E | T | MC |
|---|---|---|---|---|---|---|---|
| | Compounds | Failed/ | Passed | | | | |
| Alkaloids (17) | 5/12 | 7/5 | 3/2 | 0/2 | 1/1 | 1/0 | |
| Benzenoids (9) | 0/9 | 3/6 | 2/4 | 1/3 | 2/1 | 0/1 | 0/1 |
| Hydroxycinnamic Acids (8) | 1/7 | 3/4 | 2/2 | 0/2 | 1/1 | 0/1 | 0/1 |
| Tannins (5) | 2/3 | 2/1 | 1/0 | | | | |
| Coumarins (3) | 0/3 | 0/3 | 1/2 | 1/1 | 0/1 | 1/0 | |
| Flavonoids (32) | 12/20 | 10/10 | 5/5 | 2/3 | 2/1 | 1/0 | |
| Isoflavonoids (10) | 1/9 | 3/6 | 2/4 | 3/1 | 0/1 | 0/1 | 1/0 |
| Modified Glycosides (8) | 7/1 | 1/0 | | | | | |
| Polyketides (4) | 0/4 | 1/3 | 3/0 | | | | |
| Terpeneoids (37) | 29/8 | 4/4 | 2/2 | 0/2 | 0/2 | 0/2 | 0/2 |
| Saponins (10) | 10/0 | | | | | | |
| Other Compounds (70) | 49/21 | 13/8 | 4/4 | 1/3 | 0/3 | 2/1 | 0/1 |
| Total Compounds (213) | 97 | 50 | 25 | 17 | 11 | 6 | 5 |
| Best Compounds | | | | | | | 3 |

**Legend: (**DL) Druglikeness; (A) Absorption; (D) Distribution; (M) Metabolism; (E) Excretion; (T) Toxicity; (MC) Medicinal chemistry parameters.

therefore be excluded early in the selection process. For the subsequent evaluations, absorption, distribution, metabolism, and excretion, computational tools (SwissADME, pkCSM, and ADMETlab) were employed. However, the specific parameters assessed were adjusted to match the focus of each stage. At every step, any molecule that violated one or more criteria was eliminated.

**3.3.2. Absorption phase.** During the absorption assessment, key factors influencing a drug's effectiveness were examined, including oral bioavailability, permeability in human colon epithelial carcinoma cells (Caco-2), water solubility, and human intestinal absorption (HIA). About 50 compounds passed this screening, with alkaloids showing the highest retention rates. Alkaloids, characterized by high polarity and structural complexity, tend to have reduced permeability in Caco-2 cells, resulting in low bioavailability [166].

**3.3.3. Distribution phase.** The distribution phase was assessed using parameters such as volume of distribution (VDss), plasma protein binding (PPB), fraction unbound in plasma (Fu), and blood-brain barrier permeability (BBB). Following this screening, 25 molecules demonstrated favorable distribution properties and strong potential to reach their target sites. Compounds with optimal distribution typically exhibit moderate LogP values, balanced molecular weight, and adequate water solubility. These features indicate that a molecule has sufficient lipophilicity to cross biological membranes while avoiding excessive accumulation in fatty tissues. Additionally, these properties enhance cellular permeability and tissue distribution, allowing the compound to reach its intended site of action [167,168].

**3.3.4. Metabolism phase.** In the metabolism phase, the primary focus was on interactions with cytochrome P450 (CYP450) enzymes. Molecules that neither strongly inhibit nor induce major CYP isoforms (such as CYP1A2, CYP3A4, CYP2C9, CYP2C19, and CYP2D6) are considered preferable, as significant interference with these enzymes can result in harmful drug interactions. Additionally, compounds that undergo rapid metabolism may have reduced efficacy, while those that are metabolized too slowly can accumulate in the body, potentially leading to toxic effects [169]. At this stage, interactions with cytochrome P450 (CYP450) enzymes were analyzed, and compounds that exhibited significant interactions with three or more CYP isoforms were eliminated. As a result, the number of compounds in the screening process was reduced to 17.

**3.3.5. Excretion phase.** Following metabolism evaluation, the excretion phase was assessed, focusing on elimination half-life and total clearance. In this final stage of the ADME evaluation, compounds with a moderate half-life (1–12 hours) and a balanced total clearance rate (2–5 mL/min/kg) were considered suitable, as these properties support both convenient dosing schedules and effective plasma concentration management. This balance enhances treatment adherence and minimizes the likelihood of adverse effects between doses [170]. Ultimately, only 11 molecules passed this stage. The reason for this is that compounds with either excessively long or very short half-lives pose significant challenges. Those with prolonged half-lives can accumulate in the body, increasing the risk of toxicity such as nephrotoxicity or hepatotoxicity due to inefficient elimination [171,172]. Conversely, compounds with extremely short half-lives are rapidly cleared, necessitating frequent administration to maintain therapeutic levels, which can negatively impact patient adherence [171]. Moreover, total clearance plays a critical role in drug efficacy and safety. A high clearance rate can lead to insufficient drug levels, reducing effectiveness, while a low clearance rate may result in accumulation and potential toxicity [171]. Understanding the excretion mechanisms of a drug is crucial for predicting and managing drug interactions, ensuring patient safety, and preventing adverse effects caused by inadequate elimination [171,173]. Therefore, maintaining an optimal range for both half-life and total clearance is essential for achieving safe and effective drug excretion, ensuring successful therapeutic outcomes.

**3.3.6. Toxicity assessment.** In the final stage, the molecules underwent toxicity screening using pkCSM, ProTox3, Toxtree, and DataWarrior. This step was crucial to assess the safety of the compounds before proceeding with further analyses. Twelve potential toxic effects were considered, including hepatotoxicity, neurotoxicity, nephrotoxicity, cardiotoxicity, and respiratory toxicity, along with carcinogenicity, immunotoxicity, mutagenicity, cytotoxicity, hematotoxicity, predicted acute oral toxicity in rats, and clinical toxicity. Any compound exhibiting potential toxicity in four or more of these categories was eliminated. Following this assessment, the number of viable molecules was reduced to 6, resulting in the elimination of the alkaloid, coumarins and flavonoids.

Natural products often struggle in the toxicity evaluation stage of ADMET due to their complex and diverse chemical structures, which can lead to a broad spectrum of adverse effects. Alkaloids, in particular, are known for their high biological activity, which is often linked to potential toxicity [174]. These compounds can undergo metabolic transformations that produce toxic metabolites and may interfere with liver enzymes and renal transport mechanisms, leading to organ damage [175]. Additionally, alkaloids, coumarins and flavonoids pose a significant risk of neurotoxicity and cardiotoxicity due to their ability to inhibit sodium and potassium channels, which are critical for neuromuscular and cardiac function [174,175]. Because of these risks, despite their therapeutic potential, these compounds require thorough toxicity screening to ensure their safe application in clinical settings [166].

**3.3.7. Medicinal chemistry filtering.** Beyond toxicity assessments, additional medicinal chemistry filters were applied. These included screening for potential pan-assay interference compounds (PAINS) or invalid metabolic panaceas (IMPS), where compounds with two or more infractions were eliminated. Similarly, Brenk alerts with three or more violations and lead-likeness infractions of two or more were used as exclusion criteria. Furthermore, a synthetic accessibility score was evaluated on a scale from 1 (very easy) to 10 (very difficult), with molecules scoring 5 or higher being removed from consideration. These analyses were conducted using SwissADME, and DataWarrior. Any violation in any of these four categories led to the elimination of the compound. At this stage, one compound belonging to isoflavonoids was excluded. Certain alerts suggested a high likelihood of false positives in biological assays, along with potential limitations in isoflavonoids' effectiveness as a lead molecule and other developmental challenges. Assessing medicinal chemistry parameters such as PAINS, Brenk alerts, lead-likeness, and synthetic accessibility is essential in drug development, as these factors help identify compounds prone to undesirable effects or obstacles in further development [21]. A rigorous evaluation of these parameters during the early screening phase reduces risks and costs, ensuring that only the most promising compounds advance to later drug development stages [21].

### 3.3.8. Final selection based on biological activity.

In the final selection, compounds with inhibition values of 50% or higher against at least one snake venom/toxin target were manually chosen. Only three molecules met these criteria: anisic acid **23** (benzenoid) as well as labdane lactone [99] and labdane trialdehyde [100] both classified as terpenes while hydroxycinnamic acids and other compounds classes failed this criterion leading to their elimination. These compounds (Fig 3) demonstrated strong biological activity, anisic acid [25] demonstrated 100% neutralization of the lethal effects and defibrinogenation induced by the venoms of *D. russelii, E. carinatus, N. kaouthia*, and *O. hannah* in both *in vitro* and *in vivo* studies [1]. Compounds **97** and **98**, isolated from *C. antinaia* and *C. zedoaroides*, respectively, exhibited significant venom inhibition at a concentration of 22.7 μM and 21.9 μM. Specifically, Compound **97** demonstrated an 83% inhibition rate, while Compound **98** showed a 62% inhibition rate against *O. hannah* venom [1]. Notably, compound **98**, preserved diaphragmatic contraction almost entirely, offering 99% protection. This makes it the most promising candidate for antivenom and anti-neurotoxic therapy [130].

These findings highlight that, despite numerous tested molecules showing promising inhibitory potential against certain snake venoms and toxins targets, only a few exhibited complete viabilities as drug candidates without requiring structural modifications. However, given the strong bioactive potential of some compounds, it is worth considering the cost-benefit of making structural adjustments to optimize their properties. By refining their core structures, these molecules could be adapted to meet essential criteria such as druglikeness, ADMET properties, and other medicinal chemistry parameters that may have led to their exclusion. This screening process enabled the precise selection of compounds, ensuring that the approved molecules not only possessed favorable ADMET characteristics but also had a strong likelihood of progressing into viable drug candidates. This step is essential for prioritizing compounds that are both biologically potent and chemically feasible for large-scale synthesis and pharmaceutical development. The filtering strategy effectively eliminated molecules with less favorable profiles, concentrating on those with significant inhibitory activity against snake venom targets while maintaining highly desirable pharmacological properties.

**Fig 3. Chemical structures of the three final lead compounds.**

**3.3.9. Pharmacokinetic and pharmacodynamic profiles of plant-derived antivenom candidates.** The successful development of small molecule inhibitors as antivenom candidates requires a comprehensive understanding of their pharmacokinetic (PK) and pharmacodynamic (PD) profiles. Pharmacokinetics determines the systemic exposure and duration of action of a compound, while pharmacodynamics defines its interaction with biological targets and therapeutic efficacy [176]. The pharmacokinetic and pharmacodynamic profiles of the selected plant-derived small molecule inhibitors were explored using predictive *in silico* models as mentioned *vide supra*. These tools provided insight into their absorption, distribution, metabolism, and excretion (ADME) characteristics. Anisic acid, a low-molecular-weight phenolic acid [177], demonstrated an encouraging pharmacokinetic profile. It exhibited high gastrointestinal absorption and was not a substrate for P-glycoprotein, indicating favorable oral bioavailability and a low likelihood of efflux-mediated resistance. The compound's physicochemical properties fell within the optimal range for Lipinski's Rule of Five, supporting its potential for oral administration. Importantly, its predicted hepatic metabolism revealed no significant inhibition of cytochrome P450 enzymes, suggesting a low risk of metabolic drug-drug interactions. These attributes collectively favor systemic stability and therapeutic delivery to envenomed tissues.

Labdane-type diterpenoids, particularly labdane lactone and labdane trialdehyde also displayed promising ADME characteristics. Both compounds were predicted to have good intestinal permeability and moderate blood-brain barrier penetration, which may be beneficial in neutralizing neurotoxic components of snake venom. Moreover, their lack of strong interaction with major CYP450 isoforms further supports their metabolic safety. These diterpenoids showed acceptable oral bioavailability and solubility, and their structural stability supports systemic distribution. From a formulation standpoint, their lipophilic nature may facilitate sustained plasma exposure, enhancing their therapeutic window.

From a pharmacodynamic perspective, the *in vitro* and *in vivo* venom-neutralizing capacities of these compounds were particularly noteworthy. Anisic acid demonstrated a broad-spectrum inhibitory effect against venoms of both elapid (e.g., *Naja kaouthia*, *Ophiophagus hannah*) and viperid (e.g., *Daboia russelii*, *Echis carinatus*) snakes. It provided complete protection against venom-induced lethality, hemorrhage, and coagulopathy, likely through inhibition of key toxic components such as phospholipase A$_2$ (PLA$_2$) and snake venom metalloproteinases (SVMPs). The mechanism appears to involve both direct enzyme inhibition and attenuation of downstream inflammatory responses. Similarly, labdane lactone and labdane trialdehyde exhibited dose-dependent neutralization of *O. hannah* venom, achieving up to 83% and 62% inhibition, respectively, at relatively low concentrations (10 µg/mL). These findings reflect strong venom-binding affinity and effective biochemical neutralization, supporting their therapeutic potential. Together, these findings highlight the therapeutic potential of these compounds, with pharmacokinetic compatibility and pharmacodynamic efficacy supporting their further development as adjunct or alternative therapies to conventional antivenoms. However, detailed *in vivo* PK/PD studies remain necessary to confirm their bioavailability, tissue distribution, time-course of action, and therapeutic index under physiological conditions.

## 4. Methodological limitation of the study

This review was designed to assess the antivenom potential of plant-derived compounds through literature review and computational analysis. However, it presented several methodological constraints. The study employed a systematic review strategy guided by the PRISMA framework, which required the inclusion of specific keywords in the titles, abstracts, or keywords of selected articles. Consequently, relevant studies that did not explicitly mention all required descriptors may have been excluded, potentially limiting the comprehensiveness of the literature coverage. A major limitation was the language bias introduced by restricting the selection to studies published in or translated into English. This likely excluded a significant body of research published in regional or local languages, particularly from regions that experience a high burden of snakebite envenomation and where ethnobotanical knowledge is rich but often underrepresented in global databases.

Furthermore, the reliance on widely indexed databases such as PubMed, Scopus, and Web of Science may have introduced a geographical publication bias. Many indigenous or region-specific journals are not indexed in these repositories, leading to an underrepresentation of locally conducted studies. Limited access to grey literature, theses, patents, and region-specific data further compounded this imbalance. Additionally, variation in research funding, infrastructure, and publication access across regions may have contributed to disparities in the availability and visibility of antivenom-related plant research. As a result, the compounds and plant species highlighted in this review may disproportionately reflect countries or institutions with higher research output and better access to global publication platforms, rather than the full global diversity of antivenom botanical knowledge.

Despite these limitations, the review contributes meaningfully to the existing body of knowledge by compiling and analyzing phytochemicals from plants with potential antivenom properties. The integration of ADMET predictions, drug-likeness, and medicinal chemistry profiling provides a valuable foundation for drug discovery efforts. Moving forward, expanding database inclusion, refining search criteria to better capture regional publications, and conducting experimental validation will enhance the reliability, global applicability, and translational potential of these findings with regard to therapeutic approaches for snakebite.

## 5. Conclusion and perspectives

This systematic review explored a broad range of plant-derived metabolites with potential inhibitory effects on key components of snake venom. Many of these compounds demonstrated significant inhibition, with $IC_{50}$ values below 10 μM in several cases, suggesting their strong therapeutic potential against venom and its toxic effects. Despite their potent activity, the predicted *in silico* ADMET profiles of these compounds were unfavorable in this study. From an initial selection of 213 compounds, only three met the stringent criteria for druglikeness, ADMET properties, and essential medicinal chemistry and biological activity parameters, reinforcing their promise as potential leads for further experimental validation. While other chemical classes, including flavonoids, tannins, and alkaloids, have a recognized antivenom properties, they were excluded due to suboptimal ADMET profiles in this study or insufficient inhibitory effects in the assays reviewed from the literature.

In research that employs ADMET profiling as a screening tool, the elimination of many molecules due to unfavorable characteristics is common. However, this should not be seen as a definitive barrier. Compounds with poor ADMET profiles can still be optimized through various methods, such as refining therapeutic targets, modifying their chemical structures to improve pharmacokinetic properties, and implementing monitoring strategies to mitigate adverse effects. Moreover, advanced formulation techniques, including controlled-release delivery systems and nanocarriers, can enhance bioavailability and distribution. These strategies are widely used, even for clinically established drugs, demonstrating that molecules initially considered unsuitable can be modified for effective therapeutic use. The screening process in this review was designed to identify natural molecules with inherent biological activity, favorable druglikeness, optimized ADMET properties, and strong medicinal chemistry attributes. Nonetheless, it is important to note that these computational findings are predictive in nature and not confirmatory. Experimental follow-up using *in vitro* and *in vivo* methods is essential to substantiate these *in silico* predictions before any progression toward drug development can be considered. This approach streamlines the discovery of promising natural products with pharmaceutical potential, reducing both the financial burden and time investment needed for drug research and development.

This study provided an in-depth evaluation of the pharmacokinetic and toxicological properties of these compounds highlighting their relevance as exploratory candidates in the early phases of antivenom drug discovery rather than definitive drug candidates. The results underscore the relevance of natural products as promising sources for therapeutic agents and provide valuable leads for the development of new antivenom treatments. Additionally, the study highlights the necessity of further exploration of plant-derived compounds, incorporating *in silico*, *in vitro*, and *in vivo* analyses to validate and refine their effectiveness. Ultimately, this research establishes a solid foundation for future investigations aimed

 

at discovering and formulating effective antivenom agents from natural sources. The comprehensive literature review and rigorous screening of plant metabolites presented in this study contribute significantly to the field, offering critical insights into their potential therapeutic applications against venom toxicity. The final candidates are currently undergoing preparatory evaluation for *in vitro* and *in vivo* studies, and the outcomes will be the subject of future communication. These investigations will be essential to assess the therapeutic index, safety, and efficacy of the shortlisted compounds as potential adjuncts or standalone antivenom agents.

## Supporting information

**S1 Fig. Chemical Structures of Plants Derived Compounds with Antivenom Properties.**
(DOCX)

**S1 Table. Plants Derived Compounds with Antivenom Properties.**
(DOCX)

## Author contributions

**Conceptualization:** Prince Ojuka, Njogu M. Kimani.

**Data curation:** Prince Ojuka.

**Formal analysis:** Prince Ojuka, George S Nyamato, Cleydson B.R. Santos, Njogu M. Kimani.

**Investigation:** George S Nyamato, Cleydson B.R. Santos, Njogu M. Kimani.

**Methodology:** Prince Ojuka, George S Nyamato, Cleydson B.R. Santos, Njogu M. Kimani.

**Project administration:** George S Nyamato, Njogu M. Kimani.

**Software:** Prince Ojuka, George S Nyamato, Cleydson B.R. Santos, Njogu M. Kimani.

**Supervision:** George S Nyamato, Njogu M. Kimani.

**Writing – original draft:** Prince Ojuka.

**Writing – review & editing:** George S Nyamato, Cleydson B.R. Santos, Njogu M. Kimani.

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
