## [Decision Letter · Decision Letter 0]

7 Jul 2025

PNTD-D-25-00699A Review and In Silico Screening of Plant-Derived Snake Venom/Toxin Inhibitors: ADMET, Drug-Likeness, and Medicinal Chemistry ProfilingPLOS Neglected Tropical DiseasesDear Dr. Kimani, Thank you for submitting your manuscript to PLOS Neglected Tropical Diseases. After careful consideration, we feel that it has merit but does not fully meet PLOS Neglected Tropical Diseases's publication criteria as it currently stands. Therefore, we invite you to submit a revised version of the manuscript that addresses the points raised during the review process. Please submit your revised manuscript within 30 days Sep 05 2025 11:59PM. If you will need more time than this to complete your revisions, please reply to this message or contact the journal office at plosntds@plos.org. Please include the following items when submitting your revised manuscript: * A rebuttal letter that responds to each point raised by the editor and reviewer(s). You should upload this letter as a separate file labeled 'Response to Reviewers '. This file does not need to include responses to any formatting updates and technical items listed in the 'Journal Requirements' section below. * A marked-up copy of your manuscript that highlights changes made to the original version. You should upload this as a separate file labeled 'Revised Manuscript with Track Changes '. * An unmarked version of your revised paper without tracked changes. You should upload this as a separate file labeled 'Manuscript '. If you would like to make changes to your financial disclosure, competing interests statement, or data availability statement, please make these updates within the submission form at the time of resubmission. Guidelines for resubmitting your figure files are available below the reviewer comments at the end of this letter. We look forward to receiving your revised manuscript. Kind regards, Kartik Sunagar, Ph.D.Guest EditorPLOS Neglected Tropical Diseases José María GutiérrezSection EditorPLOS Neglected Tropical Diseases

Shaden Kamhawi

co-Editor-in-Chief

Paul Brindley

co-Editor-in-Chief

**Journal Requirements:**

At this stage, the following Authors/Authors require contributions: Prince Ojuka, George S Nyamato, Lucy Ochola, George Omondi, Cleydson B.R. Santos, and Njogu M. Kimani. Please ensure that the full contributions of each author are acknowledged in the "Add/Edit/Remove Authors" section of our submission form.

Potential Copyright Issues:

- GRAPHICAL ABSTRACT. Please confirm whether you drew the images / clip-art within the figure panels by hand. If you did not draw the images, please provide (a) a link to the source of the images or icons and their license / terms of use; or (b) written permission from the copyright holder to publish the images or icons under our CC BY 4.0 license. Alternatively, you may replace the images with open source alternatives. See these open source resources you may use to replace images / clip-art:

**Reviewers' comments:** Reviewer's Responses to Questions

**Key Review Criteria Required for Acceptance?**

**Methods**

-Are the objectives of the study clearly articulated with a clear testable hypothesis stated?

-Is the study design appropriate to address the stated objectives?

-Is the population clearly described and appropriate for the hypothesis being tested?

-Is the sample size sufficient to ensure adequate power to address the hypothesis being tested?

-Were correct statistical analysis used to support conclusions?

-Are there concerns about ethical or regulatory requirements being met?

Reviewer #1: This manuscript presents a comprehensive in silico evaluation of plant-derived compounds for potential antivenom application. The authors conducted an extensive literature review to curate a library of over 200 phytochemicals, which were subsequently screened using ADMET (Absorption, Distribution, Metabolism, Excretion, and Toxicity) profiling tools. The study is well-structured and addresses an important area in drug discovery, particularly for underserved conditions like snake envenomation.

The strength of the manuscript lies in its methodical approach to compound selection and its rigorous computational assessment. The use of drug-likeness filters and ADMET parameters provides a rational basis for identifying lead candidates. Notably, only three compounds met the criteria for favorable pharmacokinetic and safety profiles, emphasizing both the selectivity and the utility of in silico tools in early-stage drug development.

Reviewer #2: Properly done.

Reviewer #3: Lack of regional diversity: Authors presented plethora of compounds from medicinal plants against snake venoms mainly from Asia and South America continents. However, there are several researches from other continents on phytocompounds against snake venom toxins that meet the criteria set by the authors. It will make a more balanced and globally representative review if studies from other regions are also included to reflect regional diversity.

**Results**

-Does the analysis presented match the analysis plan?

-Are the results clearly and completely presented?

-Are the figures (Tables, Images) of sufficient quality for clarity?

Reviewer #1: The authors evaluated over 200 plant compounds using established ADMET profiling tools. The computational analysis filtered out the majority of compounds due to poor pharmacokinetic or toxicity parameters. Ultimately, only three compounds met the threshold for drug-likeness, with acceptable properties for absorption, metabolism, and toxicity. These compounds showed favorable bioavailability and pharmacological potential based on the in silico models employed. The table needs improvement so that it appears professional like.

Reviewer #2: Appropriately presented.

Reviewer #3: Mechanistic organization: Toxins and their inhibitors should be discussed based on molecular targets (e.g., PLA2, SVMPs, neurotoxins) to enhance clarity and mechanistic insight. Also, the underlying mechanisms, pharmacokinetics, and pharmacodynamics were not discussed and these are highlighted by authors in the justification as the major gaps in previous reviews

**Conclusions**

-Are the conclusions supported by the data presented?

-Are the limitations of analysis clearly described?

-Do the authors discuss how these data can be helpful to advance our understanding of the topic under study?

-Is public health relevance addressed?

Reviewer #1: The study demonstrates that rigorous computational screening can effectively narrow down large libraries of natural products to a small number of promising drug-like candidates. The identification of three phytochemicals with favorable ADMET profiles supports the feasibility of plant-derived inhibitors as leads for antivenom therapy. The authors rightly suggest that these compounds should undergo further in vitro and in vivo validation to confirm their therapeutic potential.

Reviewer #2: Needs minor modifications.

Reviewer #3: In silico overinterpretation: The assertion that some compounds are ready for drug development is premature and should be rephrased. Authors should emphasize the predictive not confirmatory nature of the computational analysis.

**Editorial and Data Presentation Modifications?**

Reviewer #1: Include a short discussion on the known or predicted biological activities of the three shortlisted compounds, especially in the context of venom toxin classes (e.g., metalloproteinases, PLA₂, neurotoxins).

A brief description of the authors’ future plans for in vitro or in vivo validation would improve the translational value of the study.

Highlight the structures of the 3 compounds in the main manuscript.

Reviewer #2: Minor revision

Reviewer #3: (No Response)

**Summary and General Comments**

Reviewer #1: This manuscript presents a rigorous and relevant computational strategy to identify drug-like phytochemicals with antivenom potential. With minor revisions, the work will be a strong contribution to the fields of toxinology and natural product drug discovery.

Reviewer #2: Prince Ojuka et al., have systematically executed the job of compiling the plant compounds exhibiting snake venom/toxins inhibiting/neutralizing property. It is an elaborate study highlighting the drug-likeness of compounds such as coumarins, benzenoids, polyketides, and isoflavanoids for treating venomous snake bite. The manuscript is well written, however may be accepted with minor modifications

Comments

f= mention the author`s affiliation.

Short title: It is not appropriate, require modification suitably.

Results and Discussion

The analysis revealed 11distinct chemical classes (Figure 2), with Terpenoids (37), Flavonoids (32), Benzenoids (9), Alkaloids (17), Isoflavonoids (10), Hydroxycinnamic Acids (8), Saponins (10), Modified Glycosides (8), Polyketides (4), Tannins (5), and coumarins (3). Additionally, other identified compounds were 70.

Comment: In addition to Figure 2, assigning the plant compounds presented in Figure S1 under the above mentioned respective categories appear more meaningful and convincing.

Conclusion

(Lines 1127 and 1128) The results underscore the 1128 relevance of natural products as promising sources for antivenom agents and provide valuable leads for the development of new antivenom treatments.

Line 1133, antivenom agents

Comment: Though the word antivenom may be used, it is better to use a different word, may be therapeutic agents to avoid confusions.

The authors have to specify in their conclusion, whether to use compounds 23, 97, and 98 independently/together in what ratios, and further, compounds alone or as auxiliary agents along with antivenom to treat fatal snake bite.

Reviewer #3: This study has the potential to make a significant contribution to venom research and natural antivenom discovery. However, substantial revisions are necessary to enhance the manuscript's scientific rigor, global balance, and structural clarity. I recommend a major revision.

PLOS authors have the option to publish the peer review history of their article (what does this mean? ). If published, this will include your full peer review and any attached files.

**Do you want your identity to be public for this peer review?** For information about this choice, including consent withdrawal, please see our Privacy Policy .

Reviewer #1: No

Reviewer #2: **Yes: ** Kemparaju Kempaiah

Reviewer #3: No

---

## [Decision Letter · Decision Letter 1]

15 Sep 2025

Dear Dr Kimani,

We are pleased to inform you that your manuscript 'A Review and In Silico Screening of Plant-Derived Snake Venom/Toxin Inhibitors: ADMET, Drug-Likeness, and Medicinal Chemistry Profiling' has been provisionally accepted for publication in PLOS Neglected Tropical Diseases.

Best regards,

Kartik Sunagar, Ph.D.

Guest Editor

José María Gutiérrez

Section Editor

Shaden Kamhawi

co-Editor-in-Chief

Paul Brindley

co-Editor-in-Chief

Reviewer #1:

Reviewer #2:

Reviewer #3:

Reviewer's Responses to Questions

**Key Review Criteria Required for Acceptance?**

**Methods**

-Are the objectives of the study clearly articulated with a clear testable hypothesis stated?

-Is the study design appropriate to address the stated objectives?

-Is the population clearly described and appropriate for the hypothesis being tested?

-Is the sample size sufficient to ensure adequate power to address the hypothesis being tested?

-Were correct statistical analysis used to support conclusions?

-Are there concerns about ethical or regulatory requirements being met?

Reviewer #1: Yes

Reviewer #2: Yes

Reviewer #3: Yes

**Results**

-Does the analysis presented match the analysis plan?

-Are the results clearly and completely presented?

-Are the figures (Tables, Images) of sufficient quality for clarity?

Reviewer #1: Yes

Reviewer #2: Yes

Reviewer #3: Yes

**Conclusions**

-Are the conclusions supported by the data presented?

-Are the limitations of analysis clearly described?

-Do the authors discuss how these data can be helpful to advance our understanding of the topic under study?

-Is public health relevance addressed?

Reviewer #1: Yes

Reviewer #2: Yes

Reviewer #3: Yes

**Editorial and Data Presentation Modifications?**

Reviewer #1: (No Response)

Reviewer #2: Notably, anisic acid

achieved 100% neutralization of lethality and defibrinogenation caused by N. kaouthia,

D. russelii, O. hannah, and E. carinatus venoms in both in vivo and in vitro studies.

Labdane lactone and labdane trialdehyde, isolated from C. antinaia and C.

zedoaroides, respectively, exhibited significant venom inhibition at a concentration of

10 µg/mL.

Comment: The above statement in the abstract indicates that all the plant components included in this study are in pure form. Thus, their molecular mass is known, therefore, it is better to present their molar concentrations instead of micrograms/ml. This may be extended to rest of the text wherever necessary.

Reviewer #3: I recommend that the manuscript be accepted for publication

**Summary and General Comments**

Reviewer #1: (No Response)

Reviewer #2: The authors have answered all my queries. So the manuscript may be accepted.

Reviewer #3: This study has the potential to make a significant contribution to venom research and natural antivenom discovery

PLOS authors have the option to publish the peer review history of their article (what does this mean? ). If published, this will include your full peer review and any attached files.

**Do you want your identity to be public for this peer review?** For information about this choice, including consent withdrawal, please see our Privacy Policy .

Reviewer #1: No

Reviewer #2: **Yes: ** Kemparaju Kempaiah

Reviewer #3: No

---

## [Editor Report · Acceptance letter]

Dear Dr. Kimani,

We are delighted to inform you that your manuscript, "A Review and In Silico Screening of Plant-Derived Snake Venom/Toxin Inhibitors: ADMET, Drug-Likeness, and Medicinal Chemistry Profiling," has been formally accepted for publication in PLOS Neglected Tropical Diseases.

Best regards,

Shaden Kamhawi

co-Editor-in-Chief

Paul Brindley

co-Editor-in-Chief
